# Reconsidering Generative Objectives For Counterfactual Reasoning

**Danni Lu**[1,*] **Chenyang Tao**[2,*], **Junya Chen**[2,3], **Fan Li**[4], **Feng Guo**[1,5], **Lawrence Carin**[2]

[1] Department of Statistics, Virginia Tech, Blacksburg, VA, USA
[2] Electrical & Computer Engineering, Duke University, Durham, NC, USA
[3] School of Mathematical Sciences, Fudan University, Shanghai, China
[4] Department of Statistical Science, Duke University, Durham, NC, USA
[5] Virginia Tech Transportation Institute, Blacksburg, VA, USA
`ludanni@vt.edu, chenyang.tao@duke.edu`

## Abstract

There has been recent interest in exploring generative goals for counterfactual reasoning, *e.g.,* individualized treatment effect (ITE) estimation. However, existing solutions often fail to address issues that are unique to causal inference, such as covariate balancing and counterfactual validation. As a step toward more flexible, scalable and accurate ITE estimation, we present a novel generative Bayesian estimation framework that integrates representation learning, adversarial matching and causal estimation. By appealing to the Robinson decomposition, we derive a reformulated variational bound that explicitly targets the causal effect estimation rather than specific predictive goals. Our procedure acknowledges the uncertainties in representation and solves a Fenchel mini-max game to resolve the representation imbalance for better counterfactual generalization, justified by new theory. The latent variable formulation enables robustness to unobservable latent confounders, extending the scope of its applicability. The proposed approach is demonstrated via an extensive set of tests against competing solutions, both under various simulation setups and to real-world datasets, with encouraging results reported.

## 1 Introduction

Inferring the individualized treatment effects from observational data is a fundamental challenge shared by many decision-making application domains, including healthcare [23], advertising [15], and policy making [44], among others. Recent advances in machine learning have motivated new causal inference methodologies inspired by modern learning perspectives, such as representation learning, adversarial training, etc.

In this work we focus on the problem of causal estimation from observational data, which differs from standard supervised learning in fundamental ways [60]. First, only partial observation of the potential outcomes, the one corresponding to the assigned intervention, can be made. The lack of counterfactual labels prohibits direct validation of the estimated CE. Second, observational studies are susceptible to selection bias due to *confounding*. In particular, some variables obfuscate causation as they affect both treatment assignment and outcome [81], and they may be latent. Without a proper confounder compensation mechanism, causal estimation can face severe bias.

To resolve this difficulty, the classical statistics literature has mainly focused on sample-based adjustment strategies, namely *matching* and *weighting*. Matching pairs units that are similar with respect to particular matching criteria [74], forming basic elements of synthetic *"randomized trials"*; weighting reassigns importance weights to each sample unit to create a *pseudo* population of better balance

[29, 46, 47]. Both approaches typically make the unconfoundedness assumption [65], assuming that there are no latent variables that affect both the outcome and the treatment assignment. To guard against model mis-specification-induced failures [63], balancing weights are often used in conjunction with outcome regression models to achieve double robustness [72]. However, these classical solutions are constantly challenged by modern datasets, characterized by features such as high dimensionality [12] and complex interactions [88], and they typically make the unconfoundedness assumption.

More recently, representation learning emerged as a new, promising alternative to approach covariate balance [48, 36]. Such schemes explicitly seek an intermediate (low-dimensional) representation that is both ($i$) predictive of the outcome [82]; and ($ii$) matched between treatment groups [34]. From a learning perspective, these two points serve to promote the generalization performance for counterfactual predictions [75]. On the flip side, causal perspectives also motivate invariant feature representation learning under general machine learning setups [7].

Recent strides in generative modeling techniques, such as the *variational auto-encoder* (VAE) [39] and the *generative adversarial network* (GAN) [24], have equipped causal estimation with new learning principles. Rather than appealing to predictive goals [82], these schemes learn stochastic rules that mimic the data generating procedure, *i.e.*, how to synthesize *realistic* counterfactuals based on observed data [87]. Such *generative causal models* typically relax model assumptions posited by standard causal estimation machinery, allowing black-box type inference using flexible learners such as deep networks. Despite their reported strong empirical performance, questions remain: ($i$) Confounding: Do we fully trust the observed confounders? ($ii$) Balancing: What if the covariates are unbalanced? ($iii$) Counterfactual validation: How to avoid over-fitting?

Notably, in-depth discussions on ($iii$), causal validation procedure, has received attention in the literature only recently, despite its paramount importance [8, 83]. The promise of a fully automated causal estimation procedure has inspired many (unreliable) heuristic proxies [73] (*e.g.*, plug-in surrogate or predictive loss) and principled evaluation strategies have only appeared quite recently. While scholarly consensus on best practice is yet to be reached [20], prominent examples from this category include influence function based causal validation [3] and rank-preserving causal cross-validation [71]. Of particular interest is the Robinson residual decomposition employed by the $R$-learner [56] and generalized causal forests [10], which construct a directly learnable objective.

Motivated by the preceding discussions, this work seeks a unified treatment that accommodates ($i$)-($iii$). We revisit the generative perspective of causal modeling, and demonstrate how explicitly accounting for balancing and counterfactual validation helps to improve causal estimation. In particular, we present a variational procedure, termed *Balancing Variational Neural Inference of Causal Effects* (BV-NICE), to address the challenges of generative learning for causal estimation. Our key contributions include: ($i$) repurposing variational inference as random feature representation learning scheme to facilitate causal estimation; ($ii$) reformulating the variational objective to better balance confounder representations between comparison groups; ($iii$) incorporating causal validation targets to scrutinize inferred causal effect. Our approach features direct modeling of causal effects, rather than the difference between the outcome models. It joints strength from distribution matching, representation learning and generative causal estimation, resulting a principled attempt that better addresses the challenges in counterfactual inference. To embrace a more holistic picture, we also cover related issues such as identifiability and establish border connections to the literature on causal discovery with the extended discussions found in our supplementary material (SM).

## 2 Preliminaries

**Problem setup**  We consider the basic setup under the potential outcome framework [69, 33]. Assume a sample of $n$ units, with unit $i$ associated with a covariate $\boldsymbol{X}_i \in \mathbb{R}^p$, a treatment indicator $T_i \in \{0, 1\}$ and potential outcomes $[Y_i(0), Y_i(1)] \in \mathbb{R}^2$. The fundamental problem of causal inference [32] is that only the outcome associated with the prescribed treatment is observed, *i.e.*, $Y_i \triangleq Y(T_i) = T_i Y_i(1) + (1 - T_i) Y_i(0)$, known as the factual data. The *individualized treatment effect* (ITE) is defined as the expected difference between outcome $\tau(\boldsymbol{x}) \triangleq \mathbb{E}[Y_i(1) - Y_i(0)|\boldsymbol{X}_i = \boldsymbol{x}]$, and our goal is to learn a generalizable model $\tau(\boldsymbol{x})$ that predicts the ITE given observed covariates $\boldsymbol{x}$. We often assume the decomposition $\tau(\boldsymbol{x}) = \mu_1(\boldsymbol{x}) - \mu_0(\boldsymbol{x})$, where $\mu_t(\boldsymbol{x}) \triangleq \mathbb{E}[Y(t)|\boldsymbol{x}], t \in \{0, 1\}$ are known as the outcome models. Another key concept in causal estimation is the *propensity score* (PS): $e(\boldsymbol{x}) \triangleq p(T = 1|\boldsymbol{x})$, *i.e.*, the conditional probability of receiving the treatment given $\boldsymbol{x}$. While the identifiability of causal effect can only be established in the average sense for observational studies, under the assumptions of unconfoundedness:$\{Y(0), Y(1)\} \perp\!\!\!\perp T|\boldsymbol{X}$, and positivity:

$p(T|\boldsymbol{X}, Y(0), Y(1)) \in (0, 1)$ [66], individualized predictions still hold promise. A typical predictive scheme minimizes the prediction loss for the factual observation, *i.e.*, $\hat{\mu} = \min_{\mu}\{\sum_i (Y_i - \mu_t(\boldsymbol{X}_i))^2\}$. Alternatively, generative schemes seek to identify a data generation procedure $p_{\theta}(\boldsymbol{x}, t, y)$ that is consistent with factual observations $\mathcal{D}_n = \{(\boldsymbol{x}_i, y_i, t_i)\}_{i=1}^n$.

**Robinson residual decomposition**   Under unconfoundedness, it is easy to verify $\mathbb{E}\left[\epsilon(T)|\boldsymbol{X}, T\right] = 0$, where $\epsilon(T) \triangleq Y(T) - (\mu_0(\boldsymbol{X}) + T\tau(\boldsymbol{X}))$ is known as the Robinson residual [64]. Denoting the conditional mean outcome as $m(\boldsymbol{x}) \triangleq \mathbb{E}[Y|\boldsymbol{x}] = \mu_0(\boldsymbol{x}) + e(\boldsymbol{x})\tau(\boldsymbol{x})$, and we can rewrite Robinson residual as $\epsilon(T) = Y(T) - m(\boldsymbol{X}) - (T - e(\boldsymbol{X}))\tau(\boldsymbol{X})$. Note that this decomposition holds for any outcome distribution, including binary outcomes. This directly motivates the $R$-learning [56] objective $\hat{\tau} = \arg\min_{\tau}\{1/n \sum_i (y_i - \tilde{m}(\boldsymbol{x}_i) - (t_i - \tilde{e}(\boldsymbol{x}_i))\tau(\boldsymbol{x}_i))^2\}$, where $\tilde{m}(\boldsymbol{x})$ and $\tilde{e}(\boldsymbol{x})$ are estimated surrogates for the mean outcome and propensity score model. Recently, many have considered the of direct modeling of CE ($\tau$) through the $R$-decomposition [91, 92, 16, 61, 10], rather than indirectly through $(\mu_0, \mu_1)$.

**Variational inference**   A general learning principle is to maximize the expectation of the log-likelihood wrt observed data, *i.e.*, $\ell(\theta) := \sum_i \log p_{\theta}(\boldsymbol{x}_i)$, which constitutes *maximum likelihood estimation* (MLE). For a latent variable model $p_{\theta}(x, z)$, we consider $x$ as an observation (*i.e.*, data) and $z$ as latent variable. The marginal likelihood $p_{\theta}(x) = \int p_{\theta}(\boldsymbol{x}, \boldsymbol{z}) \, dz$ typically does not have a closed-form expression, and to avoid direct numerical estimation of $p_{\theta}(\boldsymbol{x})$, variational inference (VI) instead optimizes a variational bound to the marginal log-likelihood $\log p_{\theta}(\boldsymbol{x})$ [14, 79]. The most popular choice is known as the *Evidence Lower Bound* (ELBO), given by

$$\text{ELBO} \triangleq \mathbb{E}_{Z \sim q_{\phi}(\boldsymbol{z}|\boldsymbol{x})}\left[\log \frac{p_{\theta}(\boldsymbol{x}, \boldsymbol{Z})}{q_{\phi}(\boldsymbol{Z}|\boldsymbol{x})}\right] \le \log p_{\theta}(\boldsymbol{x}), \tag{1}$$

where $q_{\phi}(\boldsymbol{z}|\boldsymbol{x})$ is an approximation to the true posterior $p_{\theta}(\boldsymbol{z}|\boldsymbol{x})$ and the inequality is a result of Jensen's inequality. This bound tightens as $q_{\phi}(\boldsymbol{z}|\boldsymbol{x})$ approaches the true posterior $p_{\theta}(\boldsymbol{z}|\boldsymbol{x})$. For estimation, we seek parameters $\theta$ that maximize the ELBO, and the commensurately learned parameters $\phi$ are often used in a subsequent inference task with new data.

**Adversarial distribution matching**   Consider the problem of matching a model distribution $p_G(x)$ to some true data distribution $p_d(x)$ presented as empirical samples, wrt some discrepancy measure, $d(p_d, p_G)$. Typically, $p_G(x)$ is given in the form of a stochastic sampler. In the GAN framework, the discrepancy is first estimated by maximizing an auxiliary variational functional $V(p_d, p_G; D) : \mathcal{P} \times \mathcal{P} \to \mathbb{R}$ between distributions $p_d(x)$ and $p_G(x)$ satisfying $d(p_d, p_G) = \max_D V(p_d, p_G; D)$, where $\mathcal{P}$ is the space of probability distributions and $V(p_d, p_G; D)$ is estimated using samples from the two distributions. Function $D(x; \omega)$, parameterized by $\omega$ and known as the *critic* function, is intended to maximally discriminate between samples of the two distributions. Subsequently, one seeks to match the generator distribution $p_G(x)$ to the unknown true distribution $p_d(x)$ by minimizing the estimated discrepancy, resulting in a minimax game between the critic and the generator: $\min_G \max_D V(p_d, p_G; D)$.

## 3   Balancing VI For Causal Estimation

Inspired by the above, we present BV-NICE, a model seeks to improve the current practice of generative learning of causal inference from the following perspectives: ($a$) automated feature representation learning that explicitly accounts covariate balance, ($b$) a built-in mechanism for automated model selection directly targets CE estimation accuracy, ($c$) acknowledging the uncertainty in the observed confounders by introduction of inferred latent variables.

We frame our construction under variational inference based on the following considerations:

- We treat covariate $\boldsymbol{x}$ as noisy proxies for the true, unobservable confounders (latent $\boldsymbol{z}$)
- The (approximate) posterior acts as a representation encoder that encapsulates uncertainties
- Matching for the prior $p(\boldsymbol{z})$ naturally regularizes for model generalization

Consider the following latent variable model $p_{\theta}(\boldsymbol{x}, y, t, \boldsymbol{z}) = p(\boldsymbol{x}|\boldsymbol{z})p(y|\boldsymbol{z}, t)p(t|\boldsymbol{z})p(\boldsymbol{z})$ (Figure S1), where $(\boldsymbol{x}, y, t)$ are the observables, $\boldsymbol{z}$ is the (continuous) latent variable, and $\theta$ denotes the model parameters. In accordance with standard practice, we model discrete variables with multinomial logistic and continuous variables with Gaussian $\mathcal{N}(\boldsymbol{\mu}, \boldsymbol{\sigma}^2)$, where $\boldsymbol{\mu}$ is a function of $\boldsymbol{z}$ and also possibly $t$ depending on the context, with $\boldsymbol{\sigma}^2$ set to some prescribed value to avoid overfitting. We parameterize stochastic encoders $q_{\phi}(\boldsymbol{z}|\boldsymbol{x}, y, t)$ to infer unobserved confounders $\boldsymbol{z}$. For flexible inference, we model all functions with deep neural nets. Plugging into (1) gives us a tractable objective

for stochastic optimization (see Equation (4)). We relegate the specifics of our modeling choices in the subsections that follow, after revealing more causal insights embodied in our reformulation.

## 3.1 A unifying view for VI and $R$-learner

A key feature we seek to incorporate is to automatically favor solutions that more accurately describe causal effect based on the factual observations. Unlike a model-selection procedure, where candidates are screened in an *ad hoc* manner, we want our model to explore the parameter space, to identify the best candidates for causal descriptives as part of training. This precludes options such as *meta-learners* [43] and influence function based estimator [3], as they function as a causal estimator and cannot be efficiently trained in an end-to-end manner. We choose to work with the Robinson residual decomposition, and show how the resulting $R$-learner [56] relates to VI. This implies our variational framework automatically assumes the model selection property.

It is convenient to denote $\mu_t(\boldsymbol{z}) \triangleq \mu_y(\boldsymbol{z}, t)$, and the causal effect estimator $\tau(\boldsymbol{z}) = \mu_y(\boldsymbol{z}, t = 1) - \mu_y(\boldsymbol{z}, t = 0)$. Under the $R$-learning framework, one models the mean outcome $m(\boldsymbol{z})$ and $\tau(\boldsymbol{z})$ rather than $(\mu_0(\boldsymbol{z}), \mu_1(\boldsymbol{z}))$. It is easy to see these two modeling choices are related by

$$
\begin{cases}
m(\boldsymbol{z}) = e(\boldsymbol{z})\mu_1(\boldsymbol{z}) + (1 - e(\boldsymbol{z}))\mu_0(\boldsymbol{z}) \\
\tau(\boldsymbol{z}) = \mu_1(\boldsymbol{z}) - \mu_0(\boldsymbol{z})
\end{cases}
\Rightarrow
\begin{cases}
\mu_0(\boldsymbol{z}) = m(\boldsymbol{z}) - e(\boldsymbol{z})\tau(\boldsymbol{z}), \\
\mu_1(\boldsymbol{z}) = m(\boldsymbol{z}) + (1 - e(\boldsymbol{z}))\tau(\boldsymbol{z}).
\end{cases}
\tag{2}
$$

A key insight is given by the observation

$$
\epsilon(\boldsymbol{z}, t, y) = y - m(\boldsymbol{z}) - (t - e(\boldsymbol{z}))\tau(\boldsymbol{z}) = y - \{t\mu_1(\boldsymbol{z}) - (1 - t)\mu_0(\boldsymbol{z})\}.
\tag{3}
$$

Note that the RHS is the residual error for prediction given $(\boldsymbol{z}, t)$. Consequently, $\ell_R(\boldsymbol{z}, y, t) = \epsilon(\boldsymbol{z}, t, y)^2 = -2\sigma^2 \log p_\theta(y|\boldsymbol{z}, t)$. Plugging this result back into the ELBO, and recalling that $p_\theta(t|\boldsymbol{z})$ is essentially the propensity score model $e(\boldsymbol{z})$, we obtain the following factorization

$$
\mathrm{ELBO}(\boldsymbol{x}, y, t | p_\theta, q_\phi) =
$$
$$
\mathbb{E}_{\boldsymbol{Z} \sim q_\phi} [\underbrace{\log p_\theta(\boldsymbol{x}|\boldsymbol{Z})}_{\text{Optional}} + \overbrace{\underbrace{\log p_\theta(y|\boldsymbol{Z}, t)}_{R\text{-loss}} + \underbrace{\log p_\theta(t|\boldsymbol{Z})}_{\text{PS-loss}}}^{\text{V-NICE}}] - \underbrace{\mathrm{KL}(q_\phi(\boldsymbol{z}|\boldsymbol{x}, y, t) \parallel p(\boldsymbol{z}))}_{\text{KL-loss}}
\tag{4}
$$

Since our primary goal is to model the causal effect $\tau$, we discard the first term related to the likelihood of $\boldsymbol{x}$ and treat the rest as our training target, which we term $\ell_{\text{V-NICE}}$. This choice is motivated by the fact that to correctly infer CE we only need the part of $\boldsymbol{x}$ that is predictive of $(y, t)$ [82]. Modeling $\boldsymbol{x}$ indiscriminately, as practiced by existing generative causal models [50, 68], takes away representation capacity of $\boldsymbol{z}$ [30, 5], compromising our main objective.

Intuitively, $\ell_{\text{V-NICE}}$, our reformulated ELBO, is a combination of $R$-loss and propensity score loss, regularized by KL-divergence on the latents to encourage better generalization. Unlike its generative counterparts, our model is directly parameterized through causal triplet $(\tau(\boldsymbol{z}), m(\boldsymbol{z}), e(\boldsymbol{z}))$ to emphasize the causal perspective and allowing structural constraints to be imposed [43]. V-NICE also approximately

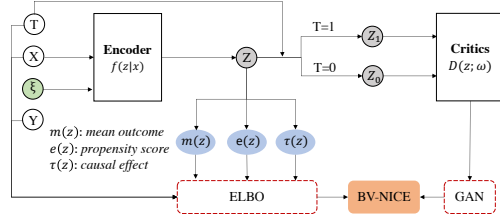

Figure 1: BV-NICE model architecture.

recovers the $R$-learner as $\sigma^2 \to 0$. By optimizing the triplet jointly, rather than a two stage procedure employed by $R$-learner, our triplet share the refined representation learned. Our discussion also bridges $R$-learning and likelihood-based learning.

**Benefits of integrating the $R$-loss.** A major difference in the construction of $R$-learner objective, relative to the standard two-learner setup, is that the propensity score is explicitly involved. This allows additional information to be leveraged in many practical settings. For example, a common scenario is that significant lags can be expected between the application of a treatment and the observation of the outcome (*e.g.*, when the target outcome is the patients' recovery in one year time whether or not administrating a drug). In such scenarios, there will be data available with only confounder and treatment to refine propensity score estimate, which in turn improves treatment effect estimation in $R$-learning, but can not be used for outcome modeling in the two learner setup. A similar argument holds when additional knowledge is known about the treatment assignment (*e.g.*, when the data is a hybrid of observational and randomized trial). In the same spirit, $R$-learning also allows the use of data where the treatment information is missing, as they can still be used to improve the estimate of average outcome $m(\boldsymbol{x})$.

## 3.2 Balancing VI for causal estimation

Our next goal is to establish a mechanism that enables covariate balance. Further denote $q_t(\boldsymbol{z}) = \int q_\phi(\boldsymbol{z}|\boldsymbol{x},t)p_d(\boldsymbol{x}|T=t)\,\mathrm{d}\boldsymbol{x}$. To achieve better balance for subsequent causal estimation, one seeks to match the confounder distributions between treatment groups, *i.e.*, $q_1$ should be close to $q_0$. To this end, we augment the original ELBO with a distribution discrepancy score $\mathbb{D}(q_0 \parallel q_1)$, resulting in

$$\ell_{\text{BV-NICE}} \triangleq \ell_{\text{V-NICE}}(p_\theta, q_\phi) - \lambda\mathbb{D}(q_0 \parallel q_1) \tag{5}$$

as our objective for balancing VI (BV-NICE), where $\lambda > 0$ specifies the regularization strength.

**Choice of discrepancy score**  While the marginal densities of $q_0$ and $q_1$ are intractable, it is relatively easy to acquire samples from them. This motivates leveraging adversarial distribution matching strategies to (indirectly) optimize the discrepancy through a *mini-max* game. Hence, we indirectly assess $\mathbb{D}(q_0 \parallel q_1)$ via the use of a critic function (the $\max$ step), and then update the model accordingly to reduce the discrepancy (the $\min$-step). In this study, we appeal to the KL-divergence as our discrepancy measure, which can be recast in its Fenchel dual form as [18, 80]

$$\mathbb{D}_{\text{KL}}(q_0 \parallel q_1) = \mathbb{E}_{q_0}\left[\log q_0 - \log q_1\right] = \max_{\nu > 0}\{\mathbb{E}_{q_0}\left[\log \nu\right] - \mathbb{E}_{q_1}\left[\nu\right] + 1\}, \tag{6}$$

and note the maximizer $\nu^*$ satisfies $\nu^* = \frac{q}{p}$. This choice is motivated by the following considerations:

- Easy implementation relative to integral probability metric (IPM)-based schemes
- It also bounds generalization performance (Sec 3.3)
- This approach also encourages parameter sharing as the ELBO involves a KL term

Note that this choice is not restrictive, as practitioners are free to choose their favorite distribution matching schemes, such as Wasserstein [75, 6], MMD [25, 48], JSD [24, 87] or other $f$-divergence [57, 78], that possess other appealing properties. See the SM for a more thorough discussion.

To implement KL-matching, we model $\log \nu$ as a deep neural network $\vartheta_\psi(\boldsymbol{z})$, as our critic function, where $\psi$ denotes the network parameters. This gives the following neural estimator for the KL term[2]

$$\hat{\mathbb{D}}_{\text{KL}}(q_0 \parallel q_1) = \max_{\psi}\{\mathbb{E}_{Z \sim q_0}\left[\vartheta_\psi(Z)\right] - \mathbb{E}_{Z' \sim q_1}\left[\exp(\vartheta_\psi(Z'))\right]\} \tag{7}$$

In our case, the distributions are characterized by a neural sampler via the reparameterization trick, *e.g.*, $q_\phi(\boldsymbol{z}|\boldsymbol{x})$ as $G_\phi(\boldsymbol{\xi}, \boldsymbol{x})$, $\boldsymbol{\xi} \sim p(\boldsymbol{\xi})$. Gradients of the sampler can be easily obtained by directly differentiating $\hat{\mathbb{D}}_{\text{KL}}$ wrt $\phi$.

## 3.3 Practical implementation

**Random feature encoder**  To enable flexible encoding of latent features, we employ a neural sampler $r_{\phi'}(\boldsymbol{z}|\boldsymbol{x})$. The $r_{\phi'}(\boldsymbol{z}|\boldsymbol{x})$ can either be explicit with a tractable likelihood [39, 40], or implicit that maps $\boldsymbol{x}$ and noise to a latent sample, *i.e.*, $\boldsymbol{z} = G_{\phi'}(\boldsymbol{\xi}, \boldsymbol{x}), \boldsymbol{\xi} \sim \mathcal{U}([-1,1]^k)$. We choose implicit feature encoder as it produces better results.

---

**Algorithm 1** BV-NICE

---
Empirical data $\hat{p}_d = \{(\boldsymbol{x}_i, y_i, t_i)\}_{i=1}^n$, imbalance $\lambda$
**for** $k = 1, 2, \cdots$ **do**
$\quad (\boldsymbol{x}, y, t) \sim \hat{p}_d, \boldsymbol{z}' \sim p(\boldsymbol{z}), \boldsymbol{z}_\phi = G_\phi(\boldsymbol{\xi}, \boldsymbol{x}), \boldsymbol{\xi} \sim p(\boldsymbol{\xi})$
$\quad \phi_{k+1} \leftarrow \nabla_\phi\{\log p_\theta(t, y|\boldsymbol{z}_\phi) - \vartheta_\psi(\boldsymbol{x}, \boldsymbol{z}_\phi) \text{ \% Encoder}$
$\qquad - \lambda[\tilde{\vartheta}_{\tilde\psi}(\boldsymbol{z}_\phi^{t=0}) - \exp(\tilde{\vartheta}_{\tilde\psi}(\boldsymbol{z}_\phi^{t=1}))]\} \text{ \% Balancing}$
$\quad \theta_{k+1} \leftarrow \nabla_\theta\{\log p_\theta(t, y|\boldsymbol{z}_\phi)\} \text{ \% Model}$
$\quad \psi_{k+1} \leftarrow \nabla_\psi\{\vartheta_\psi(\boldsymbol{x}, \boldsymbol{z}_\phi) - \exp(\vartheta_\psi(\boldsymbol{x}, \boldsymbol{z}'))\} \text{ \% Critic}$
$\quad \tilde{\psi}_{k+1} \leftarrow \nabla_{\tilde\psi}\{\tilde{\vartheta}_{\tilde\psi}(\boldsymbol{z}_\phi^{t=0}) - \exp(\tilde{\vartheta}_{\tilde\psi}(\boldsymbol{z}_\phi^{t=1}))\} \text{ \% Critic}$
**end for**

---

Another empirical decision is whether to include treatment and outcome in the encoder. Both choices induce a valid lower bound. While the inclusion is practiced in Louizos et al. [50], we argue otherwise. First, it complicates inference procedure and introduces additional approximation error, as auxiliary models must be introduced to sample the latent. Second, the casual effect identification requires that the assignment is independent with potential outcomes conditional on the covariates. The inclusion of outcome in the encoder will, on the contrary, potentially introduce bias and violates the unconfoundedness assumption [70, 77].

**Practical variants**  Modifications to the original VI procedure are often considered by practitioners for better performance, as compensation mechanism to correct for potential model mis-specification. We consider two variants that are more principally derived: $\beta$-VAE [30] and AAE [54]. The former seeks to address the potential vanishing KL, while the later explicitly targets the mismatch between the aggregated posterior and prior. Both strategies diminishes the role of KL term in ELBO, which often compromises empirical performance via synthesizing uninformative latents to reduce the mismatch to the prior. Implementation details are included in SM.

[2]Note that we have dropped the constant term for clarity.

**Inferring causal effects** Given a new observation $\boldsymbol{x}$, we wish to infer the expected effect $\tau(\boldsymbol{x})$ for a given intervention under the learned model. Since under BV-NICE causal effect $\tau(\boldsymbol{z})$ is defined based on the latent variable $\boldsymbol{z}$ rather than the observed $\boldsymbol{x}$, the estimation of the causal effect becomes a two-stage process. In the first stage we infer the hidden $\boldsymbol{z}$ given $\boldsymbol{x}$, and in the second stage we average over the latent variables to estimate the causal effect for $\boldsymbol{x}$. An estimate of the causal effect is given by $\tau(\boldsymbol{x}) \approx \frac{1}{m} \sum_j \tau(\boldsymbol{z}'_j), \boldsymbol{z}'_j \sim q_\phi(\boldsymbol{z}|\boldsymbol{x})$.

**Counterfactual cross-validation with $R$-residual.** A major obstacle in counterfactual reasoning is that due to the absence of counterfactual observations, models can not be validated directly. In our setting, we applied $R$-loss to hold out factual observations to cross-validation our model. Although it may seem similar to the CV applied in standard machine learning, a key distinction should be noted: that our CV target is explicitly defined wrt the counterfactual estimates. As noted in Nie and Wager [56], Schuler et al. [73], factual residual does not effectively assess counterfactual performance, resulting biased or unreliable estimation.

### 3.4 Generalization bounds for BV-NICE

We provide theoretical justification for the use of KL balancing. In particular, we show that the counterfactual generalization error can be bounded by the factual error plus a KL-term of the representation distributions between the treatment groups, adjusted by the variance of the conditional outcome model. We also provide additional discussions on other theoretical aspects in the SM.

**Definition 3.1.** The expected loss for the unit and treatment pair $(\boldsymbol{z}, t)$ is

$$\ell_h(\boldsymbol{z}, t) = \int_{\mathcal{Y}} L(Y_t, h(\boldsymbol{z}, t))) p(Y_t|\boldsymbol{z}) \, \mathrm{d}Y_t, \tag{8}$$

where $L(y; h)$ denotes some loss wrt observation $y$ and hypothesis $h$, and $\boldsymbol{z}$ is parameterized via the stochastic encoder $q_\phi(\boldsymbol{z}|\boldsymbol{x})$. The expected factual and counter factual losses of $h$ and $\phi$ are:

$$\epsilon_{\mathrm{F}}(h, \phi) \triangleq \int_{\mathcal{Z} \times \{0,1\}} \ell_h(\boldsymbol{z}, t) p_\phi(\boldsymbol{z}, t) \, \mathrm{d}\boldsymbol{z} \, \mathrm{d}t, \ \ \epsilon_{\mathrm{CF}}(h, \phi) \triangleq \int_{\mathcal{Z} \times \{0,1\}} \ell_h(\boldsymbol{z}, t) p_\phi(\boldsymbol{z}, 1-t) \, \mathrm{d}\boldsymbol{z} \, \mathrm{d}t, \tag{9}$$

where $p_\phi(\boldsymbol{z}, t) = \int q_\phi(\boldsymbol{z}|\boldsymbol{x}, y, t) p_d(\boldsymbol{x}, y, t) \, \mathrm{d}\boldsymbol{x} \, \mathrm{d}y$. The expected factual treated $(t = 1)$ and control $(t = 0)$ losses are

$$\epsilon_{\mathrm{F}}^{t=1}(h, \phi) \triangleq \int_{\mathcal{Z}} \ell_{h,\phi}(\boldsymbol{z}, 1) q_1(\boldsymbol{z}) \, \mathrm{d}\boldsymbol{z}, \quad \epsilon_{\mathrm{F}}^{t=0}(h, \phi) \triangleq \int_{\mathcal{Z}} \ell_{h,\phi}(\boldsymbol{z}, 0) q_0(\boldsymbol{z}) \, \mathrm{d}\boldsymbol{z} \tag{10}$$

where $q_t(\boldsymbol{z})$ is the aggregated approximate posterior of $\boldsymbol{z}$ given $t$ defined as in Sec 3.2

**Definition 3.2.** *Precision of estimating heterogeneous effects* (PEHE) for a causal effect estimator $\hat{\tau}$ is defined as $\epsilon_{PEHE}(\hat{\tau}) \triangleq \mathbb{E}\|\hat{\tau} - \tau\|_{L^2(\mathbb{P})}^2$, where $L^2(\mathbb{P})$ is the $L^2$ norm wrt feature density $\mathbb{P}(\boldsymbol{x})$.

The following statements assert the generalization error for PEHE can be bounded by the factual error plus a KL-discrepancy term, adjusted by the variance of outcome.

**Lemma 3.3.** Let $q_t, t \in \{0, 1\}$ be the marginal aggregated approximate posterior distributions defined as in Sec 3.2, $u \triangleq p(T = 1)$ is the prevalence of treatment, and $h : \mathcal{R} \times \{0, 1\} \to \mathcal{Y}$ is a hypothesis. Assume $\|l_h(\boldsymbol{z}, t)\|_\infty \leq M$ for $t = \{0, 1\}$. Then we have

$$\epsilon_{CF}(h, \phi) \leq (1 - u) \cdot \epsilon_F^{t=1}(h, \phi) + u \cdot \epsilon_F^{t=0}(h, \phi) + \frac{1}{2M} \sqrt{\frac{1}{2} \mathbb{D}_{KL}(q_0 \| q_1)} \tag{11}$$

**Theorem 3.4.** *Under the conditions of Lemma 3.3, and assuming the loss $L$ defines $l_{h,\Phi}$ is the squared loss $L(y, y') = (y - y')^2$, and define $\sigma_Y \triangleq \max_{t \in \{0,1\}} \mathbb{E}_{\boldsymbol{Z}}[(Y(t) - \mathbb{E}[Y(t)|\boldsymbol{Z}])^2]$, we have:*

$$\epsilon_{PEHE}(h, \Phi) \leq 2\epsilon_F^{t=0}(h, \Phi) + 2\epsilon_F^{t=1}(h, \Phi) + \frac{1}{M} \sqrt{\frac{1}{2} \mathbb{D}_{KL}(q_0 \| q_1)} - 4\sigma_Y^2 \tag{12}$$

This result bears resemblance to the generalization bound proved in Shalit et al. [75]. The key difference is that we have replaced the IPM bound with a KL bound. The original implementation of CFR used the Sinkhorn iterations or MMD computed their IPM, which scales quadratically wrt mini-batch size. Our Fenchel dual KL estimation scales linearly wrt sample size, and consequently more scalable. And the new assumption on $\ell_{h,\phi} \in L^\infty$ is generally easily satisfied in practice, while the RKHS assumed by CFR is difficult to verify.

# 4 Related Work

**Bayesian causal estimation** can be classified based on how the uncertainty is accounted for. Classical approaches place uncertainty on the model itself, with prominent examples such as BART [17]. To flexibly model the complex causation, Bayesian nonparametric (BNP) schemes have become popular [31]. Alaa and van der Schaar [4] investigated the fundamental limit of information rate for BNP causal models. Closest to this paper is the work of causal estimation VAE (CE-VAE) [50], where latent variables are introduced to account for the uncertainty, with the model learned through variational Bayesian analysis. Our work enhances CE-VAE by infusing additional causal perspectives into its construction: we explicitly address the covariate balancing issue and elaborate how VI connects to $R$-learning, based on which a reformulated ELBO is derived. Also highly relevant are the works of Bayesian *counterfactual risk minimization* (CRM) [85, 49], where KL-divergence on the policy (model) distributions is regularized to upper bound excess risk. Our BV-NICE differs in promoting representation balance to reduce generalization risk.

**Representation learning** has drawn considerable attention in counterfactual inference. Early work explored the use of shrinkage estimators, such as LASSO [12] and elastic-net [9]. Recently, nonlinear representation learning has gained considerable momentum in recognition of growing data complexity [48]. Popular strategies include kernelization [48], neural encoding [37], and representation embedding [82]. While most approaches adopt a deterministic design [48, 75, 2], stochastic variants are considered in the works of CE-VAE [50], CE-GAN [45] and CE-IB [59], which enable additional flexibility and better matching, and consequently improved generalization [39]. Distinct from prior arts, BV-NICE directly targets representations for causal estimation and balancing rather than focusing on predictive performance [56]. See the SM for further discussions and causal perspectives on invariant representation learning [89, 90, 7].

**Generative causal learning** is an emerging subject in causal inference. The burgeoning field of generative modeling provides ample new tools and inspiration for causal modeling. GAN-based variants have been most successful in finding direct applications for counterfactual practice [1, 42, 58, 87, 11], and to a lesser extent with variational schemes [50, 59, 62]. Indirectly, the counterfactual literature has also greatly benefited from borrowing tools originally developed for generative modeling [75], such as distribution matching schemes [25, 6]. Our work presents a principled attempt to integrate generative and causal views, by bringing together counterfactual reasoning, variational learning and adversarial matching.

**Covariate balancing** is challenged by the fragility of conventional schemes applied to modern datasets. As discussed previously, matching criteria often fail in the presence of nuisance noise [52], while the use of weighting strategies are limited by their restrictive linear assumptions [9], unreliable propensity estimates [37], or unscalable numerical schemes [29]. This motivates a variety of work exploring representation learning with direct regularization of imbalance metrics, such as Mahalanobis, Wasserstein, and MMD measures [9, 13, 93, 94], to learn a proper representation, and possibly in conjunction with a (learned) weighting strategy [37, 35], to mitigate the representation mismatch. A generalization argument was provided by Shalit et al. [75] to support such practice. While some works demonstrate the gains from adopting a sophisticated balancing criteria [86], here we advocate the use of a simple, flexible KL-balancing under a generative framework.

**Hidden confounding** is detrimental to many representation learning and covariate balancing methods that posit the *ignorability or unconfoundedness* assumption [65]. The residual confounding due to noisy measurement and unobserved confounders remains as major challenges in practice, threatening the validity of causal estimation [26]. Sensitivity analysis is advised to assess the potential effect of unmeasured confounders on causal estimates [67, 22]. Extensive investigations have been done on robust recovery of (equivalent) causal graphs with the presence of unobserved latents [53, 76, 84], and potential synergies can be exploited between recent advances in causal discovery and counterfactual reasoning. Limited by space, we defer an extended discussion on this to the SM.

**Consistency and identifiability** are key concepts of parallel interest to generalizability. Beyond the common assumption of strong ignorability, conditions to ensure identifiability in the presence of latent variables have been adequately discussed in D'Amour [21], Miao et al. [55] and the references therein, and we note their settings are drastically simpler than what's assumed by BV-NICE. Most related to this work are the emerging theories on the identifiability for latent variables under the general framework assumed by variational inference [38]. While a full exposition on the topic in the context of causal inference is beyond the scope of this study, we refer readers to our SM for some preliminary discussions.

# 5 Experiments

We consider a wide range of semi-synthetic and real-world tasks to validate our models experimentally. Details of the experimental setup are described the SM, and our code is available from `https://github.com/DannieLu/BV-NICE`. Importantly, we want to experimentally unveil aspects that are important for the design of generative causal models. More analyses can be found in the SM.

## 5.1 Experimental setups

**Model architecture, hyper-parameter tuning and data pre-processing** For all instantiations, we use fully-connected multi-layer perceptrons (MLP) as our flexible learner. We randomly sample model architectures (number of layers, hidden units) and other hyper-parameters (learning rate,

Table 1: Comparison of performance on semi-synthetic datasets

| $\sqrt{\epsilon_{PEHE}}$ | IHDP1000 | | ACIC2016 | |
| --- | --- | --- | --- | --- |
| | IN-SAMPLE | OUT-SAMPLE | IN-SAMPLE | OUT-SAMPLE |
| OLS | $0.29 \pm .09$ | $0.30 \pm .11$ | $0.52 \pm .13$ | $0.65 \pm .16$ |
| CFR | $1.47 \pm .35$ | $1.46 \pm .36$ | $0.52 \pm .14$ | $0.90 \pm .26$ |
| BART | $0.30 \pm .08$ | $0.33 \pm .11$ | $0.58 \pm .12$ | $0.70 \pm .17$ |
| CAUSAL RF | $0.63 \pm .24$ | $0.63 \pm .26$ | $0.68 \pm .32$ | $0.81 \pm .40$ |
| BV-NICE | $\mathbf{0.20 \pm .04}$ | $\mathbf{0.20 \pm .06}$ | $\mathbf{0.50 \pm .13}$ | $\mathbf{0.62 \pm .17}$ |

batch-size, regularization strength, etc.). For practical cross-validation, we use $7/3$ split for training and validation respectively, and rely on validation outcome RMSE to set best configuration [3].

**Datasets** To extensively validate the proposed procedure in a realistic setup, we consider the following four datasets: $(i)$ *IHDP1000* [31]: a semi-synthetic dataset with $1,000$ simulations of different treatment and outcomes mechanism. $(ii)$ *ACIC2016* [20]: a benchmark dataset released by *Atlantic Causal Inference Competition*, which involves 77 semi-synthetic datasets with 100 replications each. $(iii)$ *JOBS* [44]: a real-world dataset with binary outcomes, a small portion of the data comes from randomized trials. $(iv)$ SHRP2 [27]: a 3-year case-cohort study of driver behavior and environmental factors at the onset of crashes and under normal driving conditions, derived from over 1 million hours of continuous video recordings. Detailed descriptions of these datasets can be found in the SM.

**Evaluation metrics** To quantitatively assess the performance of competing causal inference procedures, we consider the following performance metrics from the literature: $(i)$, ITE accuracy as quantified by $\epsilon_{PEHE}$; $(ii)$ *policy risk* $R_{pol} \triangleq 1 - \pi_f \cdot \mathbb{E}[Y(1)|f(\boldsymbol{X}) = 1)] - (1 - \pi_f) \cdot \mathbb{E}[Y(1)|f(\boldsymbol{X}) = 0)]$ [75], where $f(\boldsymbol{x}) : \mathcal{X} \to \{0, 1\}$ denotes a decision rule whether to apply the treatment and $\pi_f$ denotes the portion of population receives the treatment under $f(\boldsymbol{x})$. Note that policy risk only applies to datasets with RCT.

**Baseline solutions** To compare, the following strong or popular causal estimation baselines are considered: linear regression (OLS, with the $T$-learner setup); *Bayesian Additive Regression Trees* (BART) [17], *Causal Random Forests* (Causal RF) [83], and *Counterfactual Regression* (CFR) [75].

## 5.2 Dissecting VI for counterfactual reasoning

We first investigate which factors greatly impact the performance to support decision choices for the construction of generative causal models. In particular, we seek answers to the following points through the lens of empirical experiments: $(a)$ level of uncertainty in feature representation; $(b)$ degree of balancing (overlapping); $(c)$ sorts of distributional regularizations.

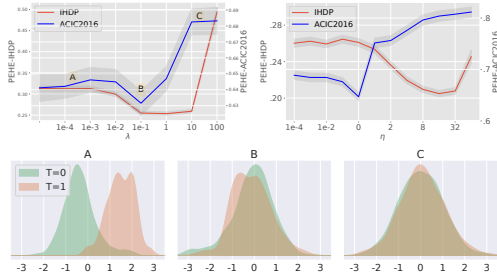

Figure 2: Impact of imbalance and randomness in feature representation. Normalized $\sqrt{\epsilon_{PEHE}}$ reported, lower is better ↓. Upper: Sensitivity to (L) imbalance $\lambda$, (R) randomness $\eta$. Lower: Projections of the under-, proper- and over-balanced feature.

To see how representation uncertainty affect performance, we introduce a randomness parameter $\eta \geq 0$, that scales the noise input to the our stochastic feature encoder, *i.e.*, $\boldsymbol{z} = G(\eta \cdot \boldsymbol{\xi}, \boldsymbol{x})$. We carried out grid search for configuration of $(\lambda, \eta)$ on both IHDP and ACIC. In Figure 2, we plot the response curves for imbalance parameter and randomness parameter, with their respective counterpart fixed at optimal. Optimal results, as measured by $\epsilon_{PEHE}$, appear at some moderate level of imbalance and representation randomness. This is consistent with theoretical predictions, because perfectly balanced representation (large $\lambda$, Fig 2C), compromise the discriminative power of latent representation, while under balanced representation (small $\lambda$, Fig 2A), are subject to the selection bias.

### 5.3 Evaluation on semi-synthetic and real datasets

Table 1 summarizes the performance of BV-NICE along with its competing solutions. For both datasets, the proposed BV-NICE performs strongly, giving best results both in terms of in-sample and out-of-sample performance. In Figure 3, we plot the mean $\sqrt{\epsilon_{PEHE}}$ computed on ACIC2016 for each simulation type. The dataset index is sorted based on out-of-sample PEHE of BV-NICE. We can see that, with very few exceptions, BV-NICE consistently outperforms its counterparts being compared. These

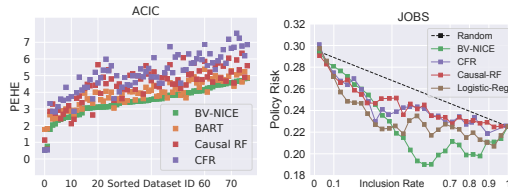

Figure 3: Result visualization on ACIC2016 (left, $\sqrt{\epsilon_{PEHE}}$) and JOBS (right, $R_{pol}$). Lower is better ↓. Index sorted for ACIC to facilitate visualization.

results underscore the importance of modeling representation uncertainty in CE estimation. Additionally, we applied BV-NICE to the JOBS dataset, and show the policy risk curve in Figure 3. In the inclusion rate regime $[0.5, 0.9]$, BV-NICE gives significantly lower risks.

### 5.4 Traffic safety risk analysis with naturalistic driving data

In our last experiment, we apply the proposed BV-NICE to analyze the risk factors in traffic safety [41, 19], in the hope that a fine-grained picture of intervention effectiveness can better inform driving safety regulations to reduce the number of tragic events. Note this study is further characterized by the challenge of rare-event modeling, due to the exceptionally low incidence rates of traffic accidents [28]. Only $1k$ crashes were flagged and annotated by trained analysts to represent the potential risk factors, along with $20k$ normal driving baselines for control. Given the prevalence of smart phone usage in modern life, our analysis concentrates on the risk analysis of cellphone use during driving.

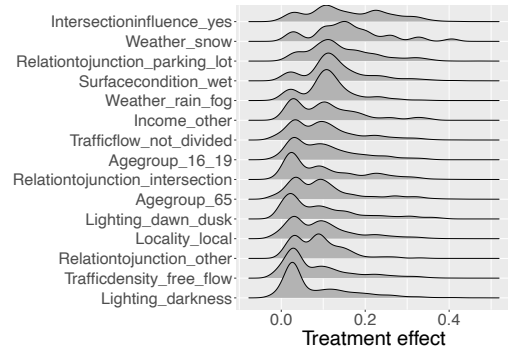

Figure 4: Cellphone risk modulation by exogenous factors, larger values imply stronger risk reduction.

Following Lu et al. [51], 11 variables are included as confounders out of 84 variables originally recorded by the study, with the inclusion criteria derived based on both domain knowledge and statistical independence tests. In Figure 4, we visualize how exogenous factors modulates the heterogenous risk distribution of cellphone use, in terms of expected reduction in incidence rate. We see restricting cellphone use is most effective in reducing collisions in bad road conditions (*e.g.*, snowy, wet, rainy, foggy), followed by complex environments (*e.g.*, parking lot crossing, intersections). More statistical summaries and comparison to alternative causal effect estimators can be found in the SM.

## 6 Conclusion

This study revisits design principles for training objective of generative causal models. In particular, we highlight the significance of covariate balancing and uncertainty of representation, which is largely missing from prior investigations. We further present a strong causal inference procedure, called BV-NICE, which bridges $R$-learning and variational inference. We extensively test our model on realistic datasets, and our results reveal the intricate nature of practical causal estimation procedures. While the empirical performance largely conforms to guiding principles, caution needs to be exercised to avoid the pitfalls, which do not appear in violation of theoretical predictions, yet can severely degrade performance. Further scrutiny is warranted for the study of robust causal estimation with flexible learners, that ameliorates the burden of exhaustive search of parameters.

## Acknowledgments and Disclosure of Funding

The authors would like to thank the anonymous reviewers for their insightful comments. The work at Virginia Tech was supported by the National Surface Transportation Safety Center for Excellence. The research at Duke University was supported was supported in part by DARPA, DOE, NIH, ONR, NSF. J Chen was partially supported by China Scholarship Council (CSC) and Shanghai Municipal Science and Technology Major Project under Grant 2018SHZDZX01 and ZJLab. The authors would also like to thank Serge Assaad, Shuxi Zeng and Wanlu Deng for fruitful discussions.

## Broader Impact

This study presents a novel generative causal inference framework, called BV-NICE, that brings together ideas from both statistical and machine learning based causal modeling. By joining the strength of variational inference, $R$-learning, and Fenchel mini-max learning, the resulting procedure fully acknowledges the representation uncertainty and enables accurate, reliable direct estimation of individualized causal effect in a flexible, scalable manner. Importantly, while there has been growing consensus that generative causal modeling such as CE-VAE is more suited for many applications yet with suboptimal performance, our research identifies the performance bottleneck and closes the gap between generative causal schemes and state-of-the-art alternatives.

This work promises to have positive societal impacts into the future. And with the best intention in the world, the author(s) wish this research will be applied to progress the course of humanity for the good. Areas stand most likely to benefit from this research are personalized healthcare, public policy, and transportation safety regulations. Variant of the proposed variational framework also promises robustness against the algorithmic biases towards the minority populations, a major issue that draws criticism for machine learning applications. This implies our model can be well suited for ensuring social justice.

## Footnotes

*Contributed equally.

[3]Note this is equivalent to the Robbinson residual validation.

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
