[Supplementary Material]

# Supplementary Material To "Reconsidering Generative Objectives For Counterfactual Reasoning"

**Danni Lu[1],[\*] Chenyang Tao[2],[\*], Junya Chen[2,3], Fan Li[4], Feng Guo[1,5], Lawrence Carin[2]**
[1] Department of Statistics, Virginia Tech, Blacksburg, VA, USA
[2] Electrical & Computer Engineering, Duke University, Durham, NC, USA
[3] School of Mathematical Sciences, Fudan University, Shanghai, China
[4] Department of Statistical Science, Duke University, Durham, NC, USA
[5] Virginia Tech Transportation Institute, Blacksburg, VA, USA
ludanni@vt.edu, chenyang.tao@duke.edu

## Contents

## A   Discussion on Causal Inference with Unobserved Confounders

Standard literature on causal inference with observational data often posits "strong ignorability", *i.e.*, all confounders are observed. This assumption is, however, untestable and oftentimes unrealistic for practical considerations. In general, estimating causal effects with unobserved confounding is infeasible [56]. Without additional assumptions, the observed data distribution can be compatible with many potentially contradictory causal explanations, that are indistinguishable based on the data in the eye of an investigator [15].

---

[\*]Contributed equally.

Figure S1: Comparison of causal graphs for different models. (Left) Standard model. (Mid) Representation learning model. (Right) Latent variable model.

One promising outlet, as adopted by many studies, is to approach a solution using *proxy variables* [47, 51]. For example, one can not directly measure the socio-economic status of a subject, and yet this might be indirectly assessed through alternative variables such as job types and shopping behavior, which are down stream of the unobserved confounding [2]. Caution that a common mispractice is to treat the proxies as if they are ordinary confounders, as bias can be expected [20, 61, 18, 57]. It has been established that causal identification is feasible under certain conditions with proxy variables are met [7, 38, 49, 65].

While the aforementioned work on proxy variables typically relies on strong assumptions to ensure identifiability under unobserved confounding, a recent trend in literature is to relax the technicality involved to enable universal algorithms for identification. These attempts often induce a latent variable setup (see Figure S1), prominent examples include [44, 70, 73, 59] [3]. Of particular popularity is the CE-VAE model [44], which is based on variational setup. Its generality and capability for modeling complex interactions with minimal modeling effort is especially significant to and well-received by emerging applications such as counterfactual reinforcement learning [5, 45]. As discussed in the main text, our BV-NICE extends and improves upon CE-VAE via inducing more causal and representation learning considerations.

Since variational inference stands as one of the most active research topics in modern machine learning [77], there exists ample opportunities to further improve VI-based causal inference procedures' performance and applicability. For example, more general variational objectives [42, 8], non-continuous unobserved confounders [27, 76, 9], and flexible priors that are data-driven [69] or encode geometric structures [52]. Our observations also echos with the recent discussion that whether tightening the bounds [60, 6] in general improves performance [58]. And from a causal perspective, proper overlapping appear to be more appropriate compared with exact balancing [78]. We will leave these questions for future investigations.

## B  Discussion on Model Generalization and Identifiability/Consistency

**Model generalization and identification of BV-NICE.**  Generalization and identification are two complementary concepts, and they are equally important, as nobody wants to generalize over unidentifiable predictions. Discussions on model identifiability in the presence of latent in the context of counterfactual reasoning have been adequately covered in the works we discussed in Section A above. Note that their setups are quite different from that of BV-NICE, which deals with more complex scenarios. Making strong, possibly unverifiable assumptions to ensure identifiability, conflicts our original intention to build an off-the-shelf causal inference framework that can deliver strong empirical performance with minimal technical assumptions. In the main text, we have presented the generalization theory for the ITE estimation of BV-NICE. For identifiability, we make the distinction of ($i$) identification of population/policy average causal effect; and ($ii$) identification of the latent variable model. The former has been addressed by the works from N. Kallus and his colleagues in the causal context [3]. For the latter, identifiability of variational inference has only been

recently studied. We give a brief review on those two streams of work in the following paragraphs respectively.

**Optimal balancing and policy evaluation.** A recent stream of work, pioneered by N. Kallus and his colleagues, focused on the robustness to unidentifiability due to unobserved confounding [35, 34, 33], framed under the evaluation of policies using observational data. Unlike the setup adopted by BV-NICE, these work seeks to construct weights that are optimal in the sense they minimize an upper bound of conditional mean square error (CMSE) of the average causal effect [30, 31, 29]. Via solving a linear constraint quadratic program (LCQP) in the reproducing kernel Hilbert space (which models the potential outcomes), such schemes optimize the worst case bound in an adversarial fashion, thereby ensuring consistency of the estimator [3]. Note that such strategies are complementary to the development of causal latent variable models such as CE-VAE, BV-NICE, or matrix completion [32]: the primary goal is the identification of the average causal effect (policy evaluation), rather than the identification of the latent variable model. In other words, such models are built on the assumption that the corresponding latent variable model can be correctly identified. There are emerging discussions on the extra assumptions needed for the identification of latent variables under the general framework assumed by variational inference (see our discussion below). And another key difference is that such policy evaluation studies focused on the ATE while BV-NICE targets the ITE.

**Identifiable variational inference and implications for causal inference.** There are emerging theories on the identifiability for the latent variables under the general framework assumed by variational inference, provided additional assumptions on weak supervision and applies to broader contexts [26, 36]. The identifiability is established in the sense that, one can recover the model parameters or the latent variables up to trivial transformations. The key difference to a standard (unidentifiable) variational auto-encoder (VAE) is that identifiable assumes additional supervision provided by observable $u$ for the latent decomposition. More formally, it seeks to learn the following conditional generative model:

$$p_{\boldsymbol{\theta}}(\boldsymbol{x}, \boldsymbol{z}|\boldsymbol{u}) = p_f(\boldsymbol{x}|\boldsymbol{z}) p_{\mathbf{T}, \boldsymbol{\lambda}}(\boldsymbol{z}|\boldsymbol{u}), \tag{1}$$

where $\boldsymbol{\theta} = (f, \mathbf{T}, \boldsymbol{\lambda})$ are the parameters, and the prior on the latent variables $p_{\boldsymbol{\theta}}(\boldsymbol{z}|\boldsymbol{u})$ is assumed to be conditionally factorial given in the following form

$$p_{\mathbf{T}, \boldsymbol{\lambda}}(\boldsymbol{z}|\boldsymbol{u}) = \prod_i \frac{Q_i(\boldsymbol{z}_i)}{Z_i(\boldsymbol{u})} \exp\left[\sum_{j=1}^{k} \mathbf{T}_{ij}(\boldsymbol{z}_i)\boldsymbol{\lambda}_{ij}(\boldsymbol{u})\right]. \tag{2}$$

And the estimation similarly optimizes the conditional ELBO. Examples of such auxiliary $u$ include temporal indices for sequence data and labels for supervised learning. The identification under equivalence class can be guaranteed under some fairly involved technical conditions, which is beyond the scope of current presentation, see [26] for details. This technique has been applied to successfully resolve the causal graph with observational data [26], and we can similarly construct identifiable BV-NICE via introducing auxiliary supervisions $u$ based on domain knowledge. We leave the exploration for future work.

## C   Discussion on the Synergies with Causal Discovery Literature

Synergies between recent advances in causal discovery and counterfactual reasoning should, and can be exploited. For many important applications, such as reinforcement learnings (RL) [45, 15], those two go hand in hand. Extensive studies have considered causal discovery in the presence of latent variables [67, 55]. This work focuses on the downstream of causal discovery: quantifying the causal effect under the assumed causal model in Figure S1. It provides a compact and flexible way to model the joint distribution of observed proxies and latent confounders, allowing further likelihood decomposition into utility driven features. To the best of our knowledge, while variational schemes have been considered in the causal discovery literature [26], our integration of balancing & the direct modeling of causal effect via Robinson factorization are novel, which have never been explored in causal discovery. Also, extension to the more general causal discovery applications will typically need non-trivial generalization of feature balancing techniques wrt non-binary interventions, a direction we seek to pursue in future work.

Another possible extension of this work that is inspired by causal discovery literature is to explicitly account for the causal sufficient set [74]. The concepts such as identification of causal sufficient adjustment sets are rarely considered in the context of counterfactual reasoning. Such techniques can help our BV-NICE framework, which practiced an explicit trial of reformulating variational inference objective to achieve feature balance, to structurally model the important covariate $X$ for engineering balanced features, and to trim sufficient features [46].

## D  Discussion on Distribution Matching Schemes

We consider three major types of distribution matching schemes: information-theoretic divergence, integral probability metric and moment matching. Below we present a brief overview, and provide discussions on their relevance to causal estimation.

**Information-theoretic divergence**  is one of the most popular metrics for evaluating distribution discrepancy. Formally, one can consider the general $f$-divergence [10], $\mathbb{D}_f(p_d \parallel p_G) \triangleq \int f\left(\frac{p_d(x)}{p_G(x)}\right) p_G(x)\, \mathrm{d}x$, where $f(\cdot) : \mathbb{R} \to \mathbb{R}$ is a convex function satisfying $f(1) = 0$, that summarizes the local discrepancy between $p_d(x)$ and $p_G(x)$. Recently, [19, 54] showed that distribution matching can be achieved via playing a mini-max game wrt the $f$-divergence. The most popular model from this category is the vanilla GAN, which optimizes Jensen Shannon divergence (JSD) and has been employed for almost all "adversarial counterfactual reasoning" procedures to date. The KL-divergence used in the proposed BV-NICE also belongs to the family, and we have embraced a more numerically stable estimation procedure through Fenchel mini-max learning [12, 68].

**Integral probability metric**  (IPM) is a popular alternative to the divergence-based objectives, as their parameter updates can be either uninformative or numerically unstable, and divergence-based objectives may not be continuous wrt the generator parameters [1]. IPM models seek to optimize an objective of the form $V_{\mathrm{IPM}}(p_d, p_G; D) = \mathbb{E}_{X \sim p_d}[D(X; \omega)] - \mathbb{E}_{X' \sim p_G}[D(X'; \omega)]$. When the critic are restricted to to Lip-1 family, IPM recovers the famous Wasserstein-1 distance, also known as the *earth mover's distance* [62]. Restricting optimization to a the Lip-1 family is a challenging task [2, 21, 50], as such many instead solve the primal form of Wasserstein using Sinkhorn-type iterations [11]. In CFR the authors have used this strategy to promote balancing [64]. One major issue with this solution is computational scalability, as quadratic computations are needed. Similar theoretical results have been derived for the benefit of IPM matching and the proposed KL-matching in the context of counterfactual reasoning. These bounds are not directly comparable and can only be assessed through experiments.

**Moment matching**  is a classical strategy employed in statistical literature to resolve the balance issues. These practice, such as entropy balancing weights and empirical likelihood, often requires solving specialized optimization problems. A modern treatment of moment matching is to leverage the reproducing kernel Hilbert space embeddings, and compute the distance btw respective embeddings $\nu_p(x) \triangleq \mathbb{E}_{X \sim p}[\kappa(x, X)]$, commonly known as the maximal mean discrepancy (MMD) $\mathrm{MMD}(p_d, p_G) \triangleq \|\nu_{p_d} - \nu_{p_G}\|_{\mathcal{H}}$, where $\mathcal{H}$ is the defining RKHS. Note that MMD is a special case of IPM and can be computed with a closed form using the kernel trick, but it also suffers the quadratic scaling like primal Wasserstein. It can be translated into an algorithm that does not require the adversarial game for generative modeling [43, 14]. Notably, MMD also belongs to the IPM family. Its use in counterfactual reasoning has also been discussed in [64]. We note that off-the-shelf MMD does not work well for high-dimensional complex data [4]. In practice, good performance can only be achieved with careful hyperparameter tuning and by introducing auxiliary loss terms to the objective [79, 40]. And such insights can be borrowed for counterfactual applications.

# E Technical Proofs

## E.1 Proof of Lemma 1

*Proof.* We have

$$
\epsilon_{CF}(h, \Phi) - [(1 - u) \cdot \epsilon_F^{t=1}(h, \Phi) - u \cdot \epsilon_F^{t=0}(h, \Phi)]
$$

$$
= \ [(1 - u) \cdot \epsilon_{CF}^{t=1}(h, \Phi) + u \cdot \epsilon_{CF}^{t=0}(h, \Phi)] - [(1 - u) \cdot \epsilon_F^{t=1}(h, \Phi) + u \cdot \epsilon_F^{t=0}(h, \Phi)]
$$

$$
= \ (1 - u) \cdot [\epsilon_{CF}^{t=1}(h, \Phi) - \epsilon_F^{t=1}(h, \Phi)] + u \cdot [\epsilon_{CF}^{t=0}(h, \Phi) - \epsilon_F^{t=0}(h, \Phi)]
$$

$$
= \ (1 - u) \int_\mathcal{X} l_{h,\Phi}(x, 1)(p^{t=0}(x) - p^{t=1}(x))dx + u \int_\mathcal{X} l_{h,\Phi}(x, 0)(p^{t=1}(x) - p^{t=0}(x))dx
$$

$$
= \ (1 - u) \int_\mathcal{R} l_{h,\Phi}(\Psi(r), 1)(q_0(r) - q_1(r))dr + u \int_\mathcal{R} l_{h,\Phi}(\Psi(r), 0)(q_1(r) - q_0(r))dr
$$

$$
\leq \ \frac{1-u}{2M} TV(q_0, q_1) + \frac{u}{2M} TV(q_1, q_0)
$$

$$
= \ \frac{1}{2M} TV(q_1, q_0) \leq \frac{1}{2M} \sqrt{\frac{1}{2} D_{KL}(q_0 \| q_1)}
$$

The last inequality is a result of the Pinsker's inequality. □

## E.2 Proof of Theorem 2

*Proof.* This result is immediate by Lemma 1 and the following inequality

$$
\epsilon_{PEHE}(h, \Phi) \leq 2\epsilon_{CF}(h, \Phi) + 2\epsilon_F(h, \Phi) - 2\sigma_Y^2, \tag{3}
$$

which is a consequence of Theorem 1 from [64]. □

# F Practical Variants of BV-NICE

$\beta$-**VAE** [23] reweighs the KL term in the ELBO term, resulting

$$
\ell_{\beta\text{-VAE}} = \frac{1}{n} \sum_i \mathbb{E}_{\boldsymbol{Z} \sim q}[\log p(\boldsymbol{x}_i | \boldsymbol{Z})] - \beta \mathrm{KL}(q(\boldsymbol{z}|\boldsymbol{x}_i) \| p(\boldsymbol{z})), \tag{4}
$$

where $\beta \geq 0$. When $\beta = 0$, it recovers the classical auto-encoder [25, 24].

**AAE** [48] instead seeks to match the aggregated posterior $q(\boldsymbol{z}) \triangleq \int q(\boldsymbol{z}|\boldsymbol{x})p_d(\boldsymbol{x})\,\mathrm{d}\boldsymbol{x}$ to the prior. Different from the original AAE, our feature encoder is stochastic, accounting for feature uncertainties. Particularly,

$$
\ell_{\mathrm{AAE}} = \frac{1}{n} \sum_i \mathbb{E}_{\boldsymbol{Z} \sim q}[\log p(\boldsymbol{x}_i | \boldsymbol{Z})] - \beta \mathrm{KL}(q(\boldsymbol{z}) \| p(\boldsymbol{z})) \tag{5}
$$

where $\beta \geq 0$. A similar model is investigated in [16] under the name *implicit VAE* (iVAE) for text generation. The general observation is that enforcing the ELBO KL term ($\mathrm{KL}(q(\boldsymbol{z}|\boldsymbol{x}) \| p(\boldsymbol{z}))$) seems to harm the performance, possibly due to over-generalization.

See Table S2 for comparisons of the above variants and other alternatives.

# G Sampling Latent Variable

An estimate of the causal effect is given by

$$
\tau(\boldsymbol{x}) \approx \frac{1}{m} \sum_j \tau(\boldsymbol{z}_j'), \tag{6}
$$

with $\{\boldsymbol{z}_j'\}_{j=1}^m$ sampled from the posterior $q_\phi(\boldsymbol{z}|\boldsymbol{x})$. This section primary focus on the case when the approximate posterior is approached in the form of $q_\phi(\boldsymbol{z}|\boldsymbol{x}, y, t)$, such that direct sampling from the (marginal) approximate posterior is infeasible given only covariate $\boldsymbol{x}$ observations [4]. The following strategies to marginalize over $(t, y)$ have been considered in this work:

Figure S2: BV-NICE performance with difference combinations of noise and imbalance levels. (Left) IHDP. (Right) ACIC. Results for IHDP copula transformed to facilitate visualization. For both schemes, best performance is achieved with moderate noise and imbalance levels.

- **MCMC** When $x$ is modeled, directly sample from $p_\theta(z|x)$ using standard MCMC. The primary concern for this strategy is computational overhead. Both gradient-based or rejection-based sampling schemes will impose considerable overhead, plus there is training effort for modeling $p_\theta(x|z)$.

- **Auxiliary outcome and propensity models** This strategy is considered by [44], where to sample $z$ given $x$, one first sample $(t', y')$ based on learned auxiliary models, then feed the synthetic pairs $(x, t', y')$ to $q_\phi$ to sample $z'$. More specifically, we partition $y$ into multiple bins, and treat it as categorical variable. We then learn $p_\phi(y|x)$ and $p_\phi(t|x)$ via standard regression. Our major concerns with this strategy is that, $(a)$ learning auxiliary model might introduce additional bias; $(b)$ sampling through the auxiliary models induces additional computation overhead.

- **Direct neural sampler** To more efficiently sample $z$ given $x$, we similarly use a neural sampler $r_{\phi'}(z|x)$, defined by $z = G_{\phi'}(\xi, x), \xi \sim \mathcal{N}(0, I)$, and train it to match the marginal distribution $q_\phi(z|x)$. We again match conditional distributions $q_\phi(z|x)$ and $r_\phi(z|x)$ wrt the KL, through a conditional critic $\vartheta_{\psi'}(z, x)$. In particular, we solve

$$\min_{\phi'} \left\{ \max_{\psi'} \left\{ \sum_i (\mathbb{E}_{Z \sim r_{\phi'}(z|x)} [\vartheta_{\psi'}(Z, x_i)] - \mathbb{E}_{Z' \sim q_\phi(z|x_i, y_i, t_i)} [\exp(\vartheta_{\psi'}(Z'_i, x_i))]) \right\} \right\}. \quad (7)$$

## H  Experimental Setups

**Neural network setup**  We use Xavier initializer for network initialization, and apply $relu$ nonlinearity except for the output layer. No batch-normalization or drop-out is used, which may further improve performance. For all critic functions, we apply an additional scaled $\tanh$ activation at the output, which greatly stabilizes training dynamics. Unless otherwise specified, we use the AAE variant of BV-NICE and fix the latent dimensions to two throughout our experiments.

The grid search parameters for the dissecting BV-NICE experiment is listed below.

- Noise level: $\eta \in \{2^k\}_{k=1}^6 + \{10^{-k}\}_{k=0}^4 + \{0\}$
- Imbalance level: $\lambda \in \{10^{-k}\}_{k=-2}^4 + \{0\}$

We show the response surface of BV-NICE wrt $(\eta, \lambda)$ in Figure S2. We found that BV-NICE is more sensitive to the injected noise intensity relative to the imbalance [5], which supports our view that properly accounting for the uncertainty merits causal inference. We remark that although in theory a neural net should be able to rescale the injected noise to its optimum regardless of $\eta$, in practice $\eta$ does regulate the level of uncertainty for the output.

**Data normalization** We found that results can be sensitive to the dynamical range of original inputs, especially for the outcome. Additionally, original PEHE can be misleading as results may be dominated by a few outliers. To ameliorate these issue, we report the normalized PEHE by rescaling the observed outcome to zero mean and unit variance based on the empirical moments of the training set. We also apply this normalization to the observed confounders as a standard pre-processing step.

While this causes some discrepancies with existing results in literature, we believe it establishes a more fair comparison btw models.

## I  Categorization of Causal Effect Learners

- $T$-**learner: Two** separate learners $(\mu_0(\boldsymbol{x}), \mu_1(\boldsymbol{x}))$ are used for the outcome prediction under different interventions.

- $S$-**learner:** A **single** learner $\mu(\boldsymbol{x}, t)$ is used for the prediction for treatment outcomes given the confounders.

- $R$-**learner:** A learner that directly models the causal effect $\tau(\boldsymbol{x})$ rather than resorting to the difference between conditional outcomes $\mu_t(\boldsymbol{x}), t \in \{0, 1\}$. Typically the Robinson decomposition is used [53].

- $X$-**learner:** Meta-learner for the causal effects via regressing on the pilot estimates produced by other causal estimation models [37]. We do not investigate the use of $X$-learner in this study.

## J  Baseline Implementations

- **(Regularized) Ordinary Least Squares (OLS):** We implemented a regularized version of OLS (*a.k.a* ridge regression) using the python *scikit-learn* package based on the $T$-learner setup. We vary the regularization strength from $10^{-5}$ to $10^3$ in $\log$ scale, at a step-size of 10 for each. The best model is picked based on cross-validation. **We found this strategy yields very reasonable performance, significantly out-performing vanilla (un-regularized) OLS and even some of the state-of-the-art alternatives.**

- **Counterfactual Regression (CFR):** Our CFR codebase is derived from `https://github.com/oddrose/cfrnet`, which includes implementations for *TARNet*, *BNN* [28] and two variants of CFR [64] (Wasserstein & MMD regularizer, rvsp). In our exploratory experiments, we find the CFR with Wasserstein regularizer usually delivers more favorable performance compared with its counterparts. As such, we choose CFR-Wasserstein as our representative from the CFR model family.

- **Causal Random Forest (Causal RF):** In [71] the authors proposed two types of Causal RF, namely *Double Sample Forest* (DSF) and *Propensity Forest* (PF). Both variants are built on the concept of "honesty", with the former leveraging non-overlapping samples for tree-construction and causal effect estimation rsvp, the latter builds the tree solely based on propensity prediction. We adopted the python implementation for both types of Causal RF from the following version of python *scikit-learn* package (`https://github.com/kjung/scikit-learn`). The PF results are reported in the main text, with the DSF results summarized in the SM here.

- **Bayesian Additive Regression Tree (BART):** We use the *bartpy* package for python BART implementation (`https://github.com/JakeColtman/bartpy`). More explicitly, the *ReadOneTrees* branch was compiled on *Jan 30, 2020* for our experiments, as we found stability issues with the *master* branch. We set the number of BART trees so that the cost of computations is comparable with other baselines.

- $R$-**learner:** Our main implementation for the $R$-learner is based on both *scikit-learn* and *tensorflow* packages. Pilot estimates of the mean outcome and propensity score models, *i.e.*, the nuisance models, are obtained by Random Forest regressor and classifier from the *scikit-learn* ensemble learner package. We use 50 trees for both cases. The causal effect $\tau(\boldsymbol{x})$ is modeled using MLP implementation in *tensorflow*, to be consistent with CFR and BV-NICE implementations. We also experimented with MLP estimator with cross-validation for the nuisance models, and found RF implementation to be more reliable.

- **GANITE:** To complement our comparison of competing solutions, we also include *GANITE*, a prominent example from the category of adversarial causal effect estimation procedures. We extracted the GANITE implementation from the python *perfect_match* package (`https://github.com/d909b/perfect_match`, [63]), and tuned its parameters for optimal performance. We report the results with unit number set to 30 and $\alpha, \beta$ set to 0.1.

- **Balancing Weights (BW):** We also experimented with explicit weighting schemes for covariate balancing. In particular, we consider propensity score based weights [41] (*e.g.*, inverse propensity weighting (IPW), matching weights (MW), overlapping weights (OW)), and entropy based (EB) weights [6] [22]. For propensity score based models, we use random forest classifier to model the propensity score. To avoid overfitting, we limit leaf size to a minimum of 5 samples. The estimated weights are supplied to a weighted regression forest to estimate $\mu(\boldsymbol{x}, t)$.

## K  Summary of Datasets

- **IHDP1000** The IHDP data comes from a study investigating the effect of specialist home visits on the cognitive test scores. This study consists of $1,000$ simulations of different treatment and outcomes mechanism. We use exactly the same data generating process in [64]. With the semi-synthetic data, we know the counterfactual outcomes $Y_i(1 - T_i)$ and the true outcome function $f_t(x)$. The dataset has 747 units with 139 treated and 608 control, 25 covariates. We use the training/test split from [64], and further hold out $30\%$ of the training samples for validation purposes.

- **ACIC2016** This is a benchmark dataset released by *Atlantic Causal Inference Competition*. It [13] involved 77 semi-synthetic datasets which shares the same covariates $\boldsymbol{X}$ but with individual outcomes and assignment mechanism $(\boldsymbol{Z}, T)$. Features were extracted from real-world observational study, while the causal mechanism are contributed by domain experts to mimic real data. This dataset include $4,802$ samples and each simulation is replicated 100 times to assess algorithmic stability.

- **JOBS** This dataset comes from a study aiming at assessing whether attending a job training program can help people get employed [39].. The data consists of a randomized study and an observational study. The dataset includes 17 covariates and 722 units, with 297 units in the treated group and 425 in the control group, from the study where units receive training randomly and 2490 control units from the PSID (Panel Study of Income Dynamics) study.

- **SHRP2** This data is derived from the Second Strategic Highway Research Plan Naturalistic Driving Study to evaluate the risk factors of car accidents. Volunteer drivers are recruited across the country, with continuous recording devices (video & telemetrics) installed to their vehicles for monitoring driving behavior in natural status. The videos of 6-second around the onset of precipitation event (5 seconds prior, 1 second after) were visually examined by trained data coders to record drivers' behavior and driving environmental variables. The normal driving segments are 6-second episodes of driving segment randomly selected from the more than 1 million hours of continuous driving videos. For those video segments, the driver demographic information, driving behavior and driving environment information were extracted.

Access to the datasets are described in `https://github.com/DannieLu/BV-NICE`. For SHRP2, data use application is needed.

## L  Additional Experimental Results and Analysis

**Importance of propensity score**   We have also observed that, even without the propensity likelihood term $p_\theta(t|\boldsymbol{z})$ in the objective, BV-NICE can still learn CE well. Using the Robinson residual alone also learns a fairly good approximation to the true propensity, and the built-in uncertainty helps to guard against over-fitting: BV-NICE based estimator remains valid approximation throughout training, while typical under-regularized learner quickly fits to the training labels. We also vary the weight of the propensity likelihood term in BV-NICE, and find that in general a small yet none-zero (*e.g.*, $\kappa = 0.01$) weight gives the best performing model.

**Learning propensity score** To see the effectiveness of Robinson decomposition in learning propensity score, we experimented with the following setups:

$$\mathcal{L} = \mathbb{E}_{\boldsymbol{Z} \sim q(\boldsymbol{z}|\boldsymbol{x})}[\rho \log p(y|\boldsymbol{Z}) + \gamma \log p(t|\boldsymbol{Z}) + \mathcal{L}_{\text{remaining}}] \tag{8}$$

Figure S3: Ablation study on propensity score for BV-NICE. It is necessary to model both propensity and outcome for accurate causal estimation.

Figure S4: Visualization of ground-truth CE versus estimated CE on the first test problem from IHDP1000 with various schemes.

- Vanilla PS: $\mathcal{L} = \log p(t|\boldsymbol{x})$
    - Direct learning of propensity score using cross-entropy with an MLP
- Outcome only: $\rho = 1, \gamma = 0$
- BV-NICE: $\rho = 1, \gamma = 1$
- Annealed BV-NICE: $\rho = 1, \gamma = 10^{-2}$

We use a flexible MLP to model the propensity score and plot the average NLL together with PEHE on both training and test set against training iterations in Figure S3 using IHDP1000. Note that while vanilla PS overfits as training progresses, BV-NICE variants showed inherent robustness against overfitting. We have also noticed that, on its own, the Robinson decomposition does not capture the propensity score well (*i.e.*, the outcome only model), and its ITE estimate is also less accurate. As such, the propensity score loss is necessary, and properly weighting it seems appropriate.

**Comparison of learned CE** In Figure S4 we compared the learned individual causal effects from different methods. Ideally the estimated CE should align along the diagonal. Performance varies, and a smaller PEHE does not always imply a good estimate based on visual inspection. In this case, our BV-NICE performs reasonably well. While some of the alternatives (CFR, Causal RF, OLS) yields estimates with tighter spread, their distributions tend to be skewed and deviate from the expected diagonal. To note, the success of $R$-learner critically relies the accuracy of (separately) learned

Table S1: Extended comparisons on semi-synthetic datasets

| $\sqrt{\epsilon_{PEHE}}$ | IHDP1000 | | ACIC2016 | |
| --- | --- | --- | --- | --- |
| | WITH-IN-SAMPLE | OUT-OF-SAMPLE | WITH-IN-SAMPLE | OUT-OF-SAMPLE |
| OLS | $0.29 \pm .09$ | $0.30 \pm .11$ | $0.52 \pm .13$ | $0.65 \pm .16$ |
| CFR | $1.47 \pm .35$ | $1.46 \pm .36$ | $0.52 \pm .14$ | $0.90 \pm .26$ |
| BART | $0.30 \pm .08$ | $0.33 \pm .11$ | $0.58 \pm .12$ | $0.70 \pm .17$ |
| CAUSAL RF (PF) | $0.63 \pm .24$ | $0.63 \pm .26$ | $0.68 \pm .32$ | $0.81 \pm .40$ |
| EB | $0.37 \pm .09$ | $0.42 \pm .16$ | $\mathbf{0.49 \pm .15}$ | $0.63 \pm .26$ |
| IPW | $0.40 \pm .12$ | $0.43 \pm .17$ | $\mathbf{0.50 \pm .15}$ | $0.64 \pm .26$ |
| CAUSAL RF (DSF) | $0.74 \pm .28$ | $0.73 \pm .30$ | $0.71 \pm .32$ | $0.84 \pm .40$ |
| GANITE | $1.04 \pm 1.04$ | $1.04 \pm 1.06$ | $0.72 \pm .33$ | $0.83 \pm .40$ |
| $R$-LEARNER | $1.27 \pm .29$ | $1.26 \pm .31$ | $0.83 \pm .12$ | $0.91 \pm .15$ |
| BV-NICE | $\mathbf{0.20 \pm .04}$ | $\mathbf{0.20 \pm .06}$ | $\mathbf{0.50 \pm .13}$ | $\mathbf{0.62 \pm .17}$ |

Figure S5: Comparison of learned representations. Note we sampled representations for BV-NICE.

Figure S6: Visualization of two unit with different treatment that are "close" in the inferred confounder distributions. Stochastic representation promotes overlapping in confounder space.

propensity. In this case, a total failure is observed. While for some of the test cases $R$-learner worked pretty well (not shown).

**Latent representations** We visualize the learned latent representation of BV-NICE and compare it with those produced from competing solutions. In Figure S5 we compare the learned distributions from TARNet, CFR and BV-NICE. The representation learned via BV-NICE shows more regularity, as instructed by the prior. To highlight BV-NICE better encourages overlapping via adopting the stochastic representation, we plot the distributions of inferred latent confounders for two units coming from opposite treatment groups in Figure S6. This confounder overlap prevents flexible learners to overfit particular units.

**Additional results on semi-synthetic datasets** Here we provide additional comparisons with baselines not summarized in the main text. Table S1 further includes comparisons with the following baselines: ($a$) GANITE, which is based on adversarial matching for causal estimation; ($b$) weighted regression, including IPW and EB weights; ($c$) vanilla $R$-learner; ($d$) double sample causal for-

Figure S7: Extended results visualization on ACIC2016. Lower is better $\downarrow$. Index sorted for ACIC based on BV-NICE to facilitate visualization.

est. Implementation details are summarized in Section J. These baselines comprehensively covers representative examples of alternative generative causal models, weighting schemes and standard $R$-learning, which complements our comparison. In Figure S7, we further compare the performance on individual test problems from the ACIC dataset, as we did in the main text.

Overall BV-NICE still performs favorably compared with the extended baselines. The only exception is that EB worked slightly better on the ACIC. We found GANITE suffered severely from stability issues, manifested by the extreme variance observed. Note that stability is a major criticism against adversarial schemes, while the Fenchel mini-max treatment adopted by BV-NICE successfully overcomes this difficulty. One possible explanation for the performance gap observed in here is that most of the causal estimation procedures we experimented do not have a proper stopping rule to guard against overfitting. For example, we experimented with the built-in stopping criteria for GANITE, and observed the results can be highly volatile. For CFR, it performed very well on the sub-problems we sampled for hyper-parameter tuning, yet showed far less satisfactory results on the full sample. We remark that the performance for some of the schemes might be limited by the particular implementation we adopted, gains might be expected with extensive recoding. However, this requirement might be too demanding for practitioners seeking off-the-shelf causal estimation solutions. Our BV-NICE highlights relatively easy implementation and robust performance.

**Comparison of BV-NICE variants** To justify our choice of BV-NICE implementation, we consider the following variants for comparison.

- Full-VAE: $q(\boldsymbol{z}|\boldsymbol{x}, y, t), \beta = 1$
- $\beta$-VAE: $q(\boldsymbol{z}|\boldsymbol{x}), \beta \in \{10^{-2}, 10^{-1}, 1\}$
- NRD-VAE: $q(\boldsymbol{z}|\boldsymbol{x})$, without Robinson decomposition
- BV-NICE: $q(\boldsymbol{z}|\boldsymbol{x})$+AAE, $\kappa \in \{10^{-2}, 10^{-1}, 1\}$

Table S2 summarizes results. The original VAE setup is observed to severely harm causal estimation.

**Summaries for NDS analysis.** In Figure S8, we compare individual causal effect estimates from CFR, BV-NICE and standard two learner. Note CFR estimates are highly concentrated, which seemed problematic given prior domain knowledge. The two learner estimates appear to be more flat compared BV-NICE estimates. We note more scrutiny is required to make final conclusions, as due to the low prevalence of crashes the validation might not be reliable, which probably explains the erratic estimates made by CFR.

**Challenges from rare-event modeling.** Dealing with severe class imbalance poses a major challenge for real-world applications, especially when the accurate classification and generalization of minority classes is of primary interest, such as traffic risk analysis discussed above. Distributions shifts are more likely for under-represented populations, thereby less generalizable with standard learning schemes. This will potentially lead to algorithmic bias: instead of capturing robust, generalizable features, a learning algorithm tend to settle with spurious correlations that is caused by the representation shift. In our setup, the representation matching scheme helps to alleviate this issue, by removing

Table S2: Comparing variants of BV-NICE.

| $\sqrt{\epsilon_{PEHE}}$ | IHDP100 | |
| | WITH-IN-SAMPLE | OUT-OF-SAMPLE |
|---|---|---|
| FULL-VAE | $1.02 \pm .38$ | $1.00 \pm .39$ |
| $\beta$-VAE ($\beta = 10^{-2}$) | $1.03 \pm .37$ | $1.01 \pm .38$ |
| $\beta$-VAE ($\beta = 10^{-1}$) | $1.05 \pm .34$ | $1.03 \pm .35$ |
| $\beta$-VAE ($\beta = 1$) | $1.03 \pm .37$ | $1.03 \pm .36$ |
| NRD-VAE | $0.26 \pm .06$ | $0.28 \pm .07$ |
| BV-NICE ($\kappa = 0$) | $0.25 \pm .07$ | $0.28 \pm .08$ |
| BV-NICE ($\kappa = 10^{-2}$) | $\mathbf{0.25 \pm .05}$ | $0.28 \pm .06$ |
| BV-NICE ($\kappa = 10^{-1}$) | $0.26 \pm .06$ | $0.28 \pm .06$ |
| BV-NICE ($\kappa = 1$) | $0.26 \pm .06$ | $\mathbf{0.27 \pm .06}$ |

Figure S8: Comparison on the NDS data.

Figure S9: NDS data summary.

apparent disparities between the distributions. And we show some preliminary success with the demonstration on the NDS driving risk analysis, and we are considering parametric assumptions that further improves the efficiency of rare-event modeling [75].. A more formal, dedicated presentation is being prepared separately.

## Acknowledgments and Disclosure of Funding

The authors would like to thank the anonymous reviewers for their insightful comments. The work at Virginia Tech was supported by the National Surface Transportation Safety Center for Excellence. The research at Duke University was supported was supported in part by DARPA, DOE, NIH, ONR, NSF. J Chen was partially supported by China Scholarship Council (CSC) and Shanghai Municipal Science and Technology Major Project under Grant 2018SHZDZX01 and ZJLab. The authors would also like to thank Serge Assaad, Shuxi Zeng and Wanlu Deng for fruitful discussions.

## Footnotes

[2]A related approach is to leverage instrumental variables to identify causal effects [66].

[3]Note that some of the practice, such as [72], has been extensively debated in literature [15], for the existence of a non-trivial ignorance region, where conflicting causal estimates can be indifferent to the investigator based on data. This renders the conclusions uncheckable and heavily dependent on modeling choices. For such cases, sensitivity analysis is recommended to reason about the causal implications [17].

[4]Note that in the main text we advocate the use of approximate posterior of the form $q_\phi(\boldsymbol{z}|\boldsymbol{x})$, which trades bound sharpness for sampling flexibility and reduced modeling bias, which turns out to be more favorable. For completeness we present discussions on sampling the infeasible posterior approximation above.

[5]This is understandable as noisy representation promotes overlapping on its own.

[6]Note EB weights ensures that the weighted first moments are matched between treatment groups.