[Reviews · NeurIPS 2020]

Review 1

Summary and Contributions: Update after rebuttal : I remain a weak accept because I"m not sure the identification with proxies is carefully discussed in the paper; modelling for uncertainty for causal inference requires specific assumptions and associated methods. Further, I think I will need to look at the further evaluation and ablations studies to judge merit of the proposed method. ============================================================== The paper presents an effect estimation method that relies on the Robinson decomposition + KL balancing to improve causal estimation. With the proposed method, both (strong) proxies of the confounder or direct confounders can be used.

Strengths: The strengths of the paper are the experimental improvements on IHDP and ACIC benchmarks. I suspect this is due to ease of KL balancing and due to the special form of the loss functions. Lemma 3.3 is a nice touch in the vein of the Johannson et al. generalization bounds.

Weaknesses: The idea of latent variable identification is very limited and should not be relied on to show causal identification directly. I think Miao et al. is a general setup of causal identification with proxies with an assumption of completeness, and even then they talk about average effects and not conditional effects. Beyond this, identification requires specific conditions and these conditions need to be discussed. I think the experiments could contain more ablation studies that show how bad estimation can get with noise in the the confounder given proxy helps or hurts. For example, increasing noise does seem to help but why? Is it just ease of training or is it something more fundamental about the data generating process? Please comment of these issues.

Correctness: The claims of robustness to unmeasured confounding are imprecise. This can only occur in certain cases like when noise in confounder is averaged out in estimates. Lemma 3.3 seems to have an issue, the supremum M should appear in the numerator. Otherwise the claim seems to be that an unbounded loss means balance does not matter.

Clarity: The paper is not very hard to read. However, the narrative in the paper can be simplified. To me, the narrative is robinson decomposition + KL balance condition improves effect estimation. Then identification relies upon identification with proxies. The latter requires more discussion.

Relation to Prior Work: The prior work is extensive but not presented in a way that is helpful and does not add insight. I think giving examples of where causal identification holds is important.

Reproducibility: Yes

Additional Feedback: The authors should clarify their discussion about identification. They should provide clear examples of where latent variable identification is possible which would then lead to causal identification. Without this, the claims in the paper are at risk of misleading readers. Please clarify the sentence "Robust to unmeasured confounding". This is a bit vague. The following is my reasoning : If your proxies determine the unobserved confounders, you immediately have effect identification. If they don't, there are conditions under which the effect is identified like when the outcome model is additive in T and Z. For a counterexample, imagine Y = T + 1[cos(k \pi Z) > 0] with large scalar k, treatment T and confounder Z, and X = Z + normal noise. This noise in Z | X would smooth out the the discontinuities which means the error in effect estimates increases with noise.


Review 2

Summary and Contributions: This paper takes a generative modelling approach towards addressing the problem of causal inference. The proposed algorithm uses the Robinson residual decomposition to derive a reformulated variational bound which is designed to explicitly estimate the causal effects rather than individual potential outcomes.

Strengths: + The claims are sound theoretically and empirical evaluations are adequate. + Causal inference is a quite relevant subject to the NeurIPS community.

Weaknesses: I have my concerns regarding the significance and novelty of this work, and I think it is not enough for publication in NeurIPS. Specifically, this work provides an improvement over a previous work, namely CEVAE [46], by adding a penalty term for learning balanced representations -- see Eq. (5). The idea of adding this penalty is not novel either, as many works (originated by [32]) have adopted and incorporated this idea into their algorithms. Moreover, the text promises to accommodate counterfactual validation in lines 62-63; however, I could not find it addressed later in the paper.

Correctness: + I acknowledge that there exists literature on estimating the causal effects directly rather than indirectly from individual potential outcomes; but I’m not familiar with it. However, I’m not sure if the proposed objective function could be optimized given the observed data; as it appears to require \tau values which are never observed. Could the authors please elaborate on how they provide the \tau(x) values for training their model? This comment also holds for m(x) as we never observe both \mu_0 and \mu_1 for the same subject. + Eq. (12): The reason why many works use integral probability metrics is that KL-divergence measure of discrepancy between two probability distributions is rather unstable numerically. Please comment on why you think this won’t be an issue with your method.

Clarity: This is a very well-written paper.

Relation to Prior Work: Yes, the related work section fully discusses how this work differs from previous contributions.

Reproducibility: Yes

Additional Feedback: + Line 147: did you mean “exclude” instead of “preclude”? + There are a couple of mistakes in Eq. (3); the correct versions are: y - m - (t - e) \tau; and y - {t \mu_1 + (1 - t) \mu_0}. Please verify whether these were only typos. + Lines 157-158: My understanding is that the factorization in Eq. (4) is a direct result of the assumed graphical model (Fig. S1.c). The authors however state that plugging the result of Eq. (3) into ELBO yields this factorization. I think what they meant to say was that in their implementation, they substitute the \tau-loss term with its \epsilon equivalent. Please clarify. ===== post-rebuttal ===== The authors have addressed many of my concerns in their rebuttal; however, I still have my concerns regarding novelty and the claim that the existing generative objectives do not account for selection bias. I have updated my score accordingly.


Review 3

Summary and Contributions: The authors use deep generative models for causal effect estimation from a Rubin causal inference perspective. The attempt to use adjust variational inference to account for balancing. They introduce a validation technique called "counterfactual validation." These two things; (1) adjusting the objective for causal constraints and estimation procedures and (2) validation procedures that incorporate causal semantics, are high impact.

Strengths: I've seen work that tries to use deep generative modeling to get parameteric identifiability under confounding (e.g. using IV) with VAEs. This is the first one I've seen that takes practical estimation concepts from the Rubin causal inference literature and uses ties it to the deep generative objective function. I also

Weaknesses: Works in this vein lack proofs of identifiability. That is a weakness here as well, though the authors do a good job addressing it.

Correctness: I found no issues with the claims.

Clarity: Yes. Extremely so.

Relation to Prior Work: I am satisfied with the discussion of prior work. Especially when it comes to identifiability.

Reproducibility: Yes

Additional Feedback: Regarding claim "And in our follow-up investigations, we have found that variant of the proposed variational framework shows robustness against the algorithmic biases towards the minority populations, a major issue that draws criticism for machine learning applications." That's waaaay to hand-wavy. Back that up in the supplement. Please don't just play lip-service to these issues. Unlike the broad ML community, causal inference research has mathematically concrete things to say about fairness/bias/discrimination.


Review 4

Summary and Contributions: This manuscript proposes a novel computational approach to ITE estimation, based on Bayesian models, variational inference and domain adaptation. First, it uses a known decomposition of the ITE to formulate the estimation as an appealing Bayesian inference problem, solved with variational inference. Second, it proposes an original penalty for dealing with unbalance, theoretically justified. Then, the authors expose a set of experiments for which their experiments outperform state-of-the-art.

Strengths: The paper is extremely comprehensive. Indeed, it might even get confusing at some points, because many topics are brought up and it is not always clear whether the paper in hand is attempting to solve them or not (more details on the clarity section). Unifying VI and the R-learner is an elegant paradigm that should be relevant to the NeurIPS community. ITE estimation is a specific flavor of counterfactual inference, it would be interesting to contextualize it for other offline counterfactual problems (batch learning from bandit feedback for example). It is an appealing point of the method that the variational formulation of the KL divergence (using the Fenchel dual form) allows for a gain in complexity, compared to Wasserstein and MMD.

Weaknesses: ITE estimation is a well established line of work. Although this paper is comprehensive and has the merit of laying out where this field may progress, as well as some contributions, there are some limitations to the novelty. Regularizing variational inference with KL divergence has a Bayesian motivation that is discussed in this recent manuscript [1]. Proposing a new regularization scheme for ITE estimation (or another counterfactual problem) along with tractable approximation is investigated in many research work. For example, [2] presents the chi-square divergence as the variance of importance sampling (in batch learning for bandit feedback) and proposes to minimize it with a variational formulation (f-divergence). [1] https://arxiv.org/pdf/1806.11500.pdf [2] http://proceedings.mlr.press/v80/wu18g.html I do not know whether this paper was intended to discuss the problem of identifying latent confounding variables (cf. clarity) but the experiment does not underline any of that. Similarly, the paper mentions briefly counterfactual cross-validation in the introduction but this is not explored (``normal’’ cross validation is used in the experiments).

Correctness: To the best of my knowledge, all the exposed claims and methods seem reasonable. I have some minor points. Line 230. The estimate of the causal effect makes use of the variational distribution. First, this can be biased. Second, VI is known to underestimate the variance in the posterior distribution. There might be some improvement with using importance weighted variational inference and using a self-normalized importance sampling estimator? Especially in scenarios where the uncertainty in tau is important. Line 248 The square root of a KL divergence usually comes from a Pinsker inequality (and it is the case here). However, the total variation distance is symmetric while KL is not. So you could get a similar bound with KL(q_1 || q_0). Why not use a symmetrized version (sum of reverse and forward KL)? Table 1: CFR has extremely poor performance. This is mentioned somewhere in the Appendix but because the datasets are the same, and that in their paper (and to my knowledge some others) CFR outperforms BART, OLS, etc.. I am intrigued. Is this something the authors kept working after the submission deadline? It would be good to know why it doesn’t work.

Clarity: The paper is compelling and well written. I have some minor comments. + Line 52-54: the way it is written, it seems like this manuscript is attempting to solve points (i), (ii) and (iii). However, (ii) only is addressed, and this is a sensibly studied problem. + The paper does not make it clear whether the latent variable z factors in hidden confounding factors, or whether it just helps in predicting the ITE and its noise, etc. + The sigma introduced in line 174 is not explained. The statement "V-NICE also approximately recovers the R-learner for sigma -> 0" is therefore not clear. Should this be added as a proof somewhere? I could not find it in the SM after a quick glance + After a bit of time, it seems like the Fenchel dual form of the KL is used twice, one for the ELBO and once for the balancing term. However, this is not explained at lines 197-204 and need to be better exposed. The notation theta prime in Alg 1 is not used anywhere else in the manuscript. + I found that the number of baselines was weak. However, I found more results in the supplements. So, there should be more links between the main text and the supplementary files. Typos: Missing a space line 11 Green in line 74 Word missing at 100 RHS not explained at 156 The list of citations must be worked through, some entries are missing and some others are poorly formatted.

Relation to Prior Work: Related work is clearly discussed in a very nice section (except the citations I proposed earlier)

Reproducibility: Yes

Additional Feedback: I would like to thank the authors for their clarification on counterfactual cross-validation and other questions I had. I maintain my score, I think this is a good paper.

[Author Response · NeurIPS 2020]

We thank the reviewers for their insightful & constructive feedback, to which we have carefully responded below.

• *(Shared by R2, R4) Clarify the claim on counterfactual cross-validation (CV).* Counterfactual CV (CF-CV) means
counterfactual term $\tau(x)$ is directly used in the CV target (our Robinson residual). In contrast, "normal" CV in ITE
would evaluate factual residual $\epsilon = Y_t - \mu_t(z)$, which does not effectively assess counterfactual performance. See [51,
67] for formal discussions & empirical evidence on how CF-CV avoids biased or unreliable estimation suffered by
normal CV. Also, there are settings with additional information on more precise propensity $e(z)$ (*e.g.*, randomized trials,
missing / pending outcomes) to improve Robinson residual based CF-CV. We have made these clear in our revision.

*Reviewer #1.*

• *Clarify "more robustness to unmeasured confounding."* We agree the statement needs to be more precise. We
advocate the view that assuming unmeasurable confounders is a more natural, robust alternative relative to assuming no
unmeasured confounders, known to be invalid for many empirical settings. (*e.g.*, in targeted advertising true confounder
should be the latent consumer profile (*e.g.*, buying power, personal flavor, etc.), not observed past purchase records
(*i.e.*, proxies)) Note endogenous uncertainties arise in data-generation, and as noted in [20], encoding proper latent
uncertainties in the model is a valid, possibly less restrictive alternative to *ad hoc* sensitivity analysis [21]. With little
scholarly consensus on the matter, we advise modeling with uncertainty unless domain knowledge suggests otherwise,
in the same vein to the fixed effects models that are commonly used in econometrics [Angrist & Pischke (2009)].

• *Comments on causal identification with proxies.* These are excellent points. The community has very different
takes on "identification" (see positive comments from R3), and improving upon Miao *et al.* remains an open challenge.
We will expand discussion on causal identifiability, listing settings & examples where it might be feasible. Informally,
for individual-level identifiability, we first need latent identifiability & verify latent effects can be averaged out.

• *More ablations to show when noise helps or hurts.* Note the noise in the proxy model $p(x|z)$ diminishes information
& leads to uncertainties in causal estimates, while the noise in the inference model $q(z|x)$ helps prevent overfitting
causal estimates until it starts to hurt training stability. More results and analyses will be added to clarify the interplay.

*Reviewer #2.*

• *Do you need values of $\tau(z)$ for training?* No, we do not. We believe the reviewer misunderstood the $R$-learning
framework, and we wish to clarify. Here $\tau(z)$ is directly trained using factual data $(x, t, y)$ using the Robinson
factorization defined in Eqn (3). The nuisance components, namely the mean outcome $m(z)$ and propensity $e(z)$, are
learned using factual pairs $(x, y)$ & $(x, t)$, respectively, with standard regression. We do not model $\mu_0, \mu_1$ in BV-NICE.

• *Novelty of the work.* Our key insight is that existing generative causal models failed to account for covariate balance
& counterfactual validation, resulting in compromised performance, and we provide a new method accounting for that.

• *Why use KL instead of the IPM metrics (e.g., Wasserstein, MMD).* IPM metrics are not without caveats. Its primal-
form estimation suffers quadratic scaling, and its dual-form estimation requires intricate constrained optimization. Our
KL estimation is simple, linear scaling, numerically stable, and it yields very strong performance.

*Reviewer #3.* We thank the reviewer for the very positive comments.

• *Backing up "BV-NICE variant alleviates algorithmic bias for minorities."* We appreciate this suggestion. Infor-
mally, distribution shifts are more likely for under-represented populations, thereby less generalizable with standard
learning schemes, but can be better handled with BV-NICE variants. We will extend the supplemental material with
clear definitions & setups, and present some results. A more formal, dedicated presentation is being prepared separately.

*Reviewer #4.*

• *Explain bad performance for some baselines, and additional results in the SM.* As documented in our experimental
setup, we fix the representation dimension to two for all representation-based models. Some baseline models (*e.g.*,
CFR) turned out to be very sensitive to the representation dimension, yielding bad results in low-dimensions. Limited
by space, we present only key results in the main text upon submission. We will use the extra page offered by the final
version to provide a more comprehensive presentation and analysis of the additional experimental results.

• *Relation to the mentioned works on regularizing causal models.* Thank you for pointing out these interesting refs
on counterfactual risk minimization (CRM), which we have carefully read and added to our discussion. We note that
their motivations, target objectives, and estimation procedures are very different from our work: these CRM models
regularize KL divergence on the policy (model) distributions to upper bound excess risk, not to promote representation
balance as in BV-NICE. Nevertheless, together they present a more holistic picture of how Bayesian formulation and
information-theoretic regularization help to improve counterfactual reasoning.

• *Does the paper intend to identify latent confounders?* We are not explicitly pursuing that goal here. Discussions on
the additional assumptions for latent confounder identification (see supplemental material) are provided for completeness,
and to bridge our future work on associating causal interpretations. The main take-away is that representation balancing,
incorporating uncertainty & direct modeling of causal effects are important factors to consider for (black-box) generative
causal modeling, which we have demonstrated with the success of BV-NICE, without enforcing latent identification.

• *Other minor comments.* We can generalize to symmetrized KL. VI variants can help. $\sigma$ is hyper-parameter for
noise & justification added. $Z$ models hidden confounding (related to identifiability). Alg 1 clarified as suggested. See
replies above on counterfactual CV & robustness to unmeasured confounding for how we address points $(i)$ and $(iii)$.

We also enriched our discussions in the broader impact statement, and fixed the typos and references.

[Meta-Review · NeurIPS 2020]

The reviewers appreciated the contribution and clarity of the paper. In the discussion the main things the reviewers wanted to see is: (a) Better clarification of the discussion on identification, with clear examples where latent variable identification is possible; (b) Some updated comparisons with CFR-WASS or other modern methods based on IHDP and ACIC; (c) An ablation study showing how each new component helps. If the authors could add these things to the final version it would further strengthen the paper. I vote to accept.